# Collapsed Variational Bounds for Bayesian Neural Networks

**Marcin B. Tomczak, Siddharth Swaroop, Andrew Y. K. Foong, Richard E. Turner**
University of Cambridge
Cambridge, UK
{mbt27,ss2163,ykf21,ret26}@cam.ac.uk

## Abstract

Recent interest in learning large variational Bayesian Neural Networks (BNNs) has been partly hampered by poor predictive performance caused by underfitting, and their performance is known to be very sensitive to the prior over weights. Current practice often fixes the prior parameters to standard values or tunes them using heuristics or cross-validation. In this paper, we treat prior parameters in a distributional way by extending the model and collapsing the variational bound with respect to their posteriors. This leads to novel and tighter Evidence Lower Bounds (ELBOs) for performing variational inference (VI) in BNNs. Our experiments show that the new bounds significantly improve the performance of Gaussian mean-field VI applied to BNNs on a variety of data sets, demonstrating that mean-field VI works well even in deep models. We also find that the tighter ELBOs can be good optimization targets for learning the hyperparameters of hierarchical priors.

## 1 Introduction

There has been a lot of recent interest in developing methods for Bayesian Neural Networks (BNNs), and variational methods provide computationally cheap approximations to the posterior over weights when compared to alternatives like MCMC. Research on Variational Inference (VI) for BNNs has seen multiple advances enabling optimization of the Evidence Lower Bound (ELBO) [29, 65, 24, 7, 38], allowing for scaling to large neural networks and datasets [57, 63, 15, 16, 66]. However, the under-confidence of mean-field VI (MF-VI) in output-space has caused poor adoption of VI in applications, especially in sequential decision making [9, 60, 69]. Although a lot of work has improved performance by deriving better estimators of the gradient of the ELBO [29, 65, 24, 7, 38], this work has focused on the *variational parameters* of distributions over weights, with the prior over weights usually fixed to zero-mean isotropic Gaussians [55, 70, 74, 15, 20, 7, 40].

In this paper, we also apply inference to the *prior parameters* of Gaussian BNNs (e.g. means and variances of a Gaussian prior over weights). We do this by using collapsed VI bounds [36, 43, 47, 64], which analytically solve the inference over prior parameters and allow us to derive tighter ELBOs for performing VI in BNNs. Many previous works argue that point estimates for parameters of Gaussian prior in BNNs can be learned reliably by maximizing the ELBO [72, 52, 50, 31], although there have also been arguments against such an approach [7]. Experiments using our derived ELBOs lead to two findings: (i) MF-VI performs well in deep models where the prior parameters of Gaussian distributions over weights have been learned, but severely under-fits data in output space if the prior over weights is fixed and, (ii) the derived tighter ELBOs are good optimization targets to learn the hyperparameters of hierarchical Gaussian priors, even in large networks. We provide the code implementing the introduced algorithms at `https://github.com/marctom/collapsed_bnns`.

35th Conference on Neural Information Processing Systems (NeurIPS 2021).

## 2 Collapsed Variational Bounds for Bayesian Neural Networks

We consider performing VI in a supervised learning scenario given $N$ data points $\mathcal{D} = \{(\boldsymbol{x}_i, \boldsymbol{y}_i)\}_{i=1}^{N}$. We denote a neural network as $f_{\boldsymbol{W}}$, the observation model as $p(\boldsymbol{y}|f_{\boldsymbol{W}}(\boldsymbol{x}))$, where $\boldsymbol{W}$ denotes the vectorized weights and biases of the network, and the likelihood of the weights $\boldsymbol{W}$ as $p(\mathcal{D}|\boldsymbol{W}) = \prod_{i=1}^{N} p(\boldsymbol{y}_i|f_{\boldsymbol{W}}(\boldsymbol{x}_i))$. We focus on Gaussian mean-field priors (with non-zero means) over vectorized network weights $p(\boldsymbol{W}|\boldsymbol{\mu}_p, \boldsymbol{\sigma}_p^2) = \mathcal{N}(\boldsymbol{W}|\boldsymbol{\mu}_p, \boldsymbol{\sigma}_p^2)$ meaning $w \sim \mathcal{N}(\mu_p, \sigma_p^2)$ for any coordinate $w$ in $\boldsymbol{W}$ (both $\boldsymbol{\mu}_p$ and $\boldsymbol{\sigma}_p^2$ are also vectors in this notation). Similarly, we restrict our attention to mean-field Gaussian approximate posteriors $q(\boldsymbol{W}|\boldsymbol{\mu}_q, \boldsymbol{\sigma}_q^2) = \mathcal{N}(\boldsymbol{W}|\boldsymbol{\mu}_q, \boldsymbol{\sigma}_q^2)$. VI finds an approximate posterior $q(\boldsymbol{W}|\boldsymbol{\mu}_q, \boldsymbol{\sigma}_q^2)$ by minimizing the KL divergence $D_{KL}(q(\boldsymbol{W}|\boldsymbol{\mu}_q, \boldsymbol{\sigma}_q^2)||p(\boldsymbol{W}|\mathcal{D}, \boldsymbol{\mu}_p, \boldsymbol{\sigma}_p^2))$ w.r.t. the variational parameters $\boldsymbol{\mu}_q, \boldsymbol{\sigma}_q^2$. This is done by optimizing the Evidence Lower Bound (ELBO) [28, 4]:

$$\mathcal{L}(\boldsymbol{\mu}_q, \boldsymbol{\sigma}_q^2; \boldsymbol{\mu}_p, \boldsymbol{\sigma}_p^2) = \sum_{i=1}^{N} \mathbb{E}_{q(\boldsymbol{W}|\boldsymbol{\mu}_q, \boldsymbol{\sigma}_q^2)} \log p(\boldsymbol{y}_i|f_{\boldsymbol{W}}(\boldsymbol{x}_i)) - D_{KL}(q(\boldsymbol{W}|\boldsymbol{\mu}_q, \boldsymbol{\sigma}_q^2)||p(\boldsymbol{W}|\boldsymbol{\mu}_p, \boldsymbol{\sigma}_p^2)). \tag{1}$$

We denote the first term on RHS as $\mathcal{L}_{data}(\boldsymbol{\mu}_q, \boldsymbol{\sigma}_q^2)$. As $D_{KL}(q(\boldsymbol{W}|\boldsymbol{\mu}_q, \boldsymbol{\sigma}_q^2)||p(\boldsymbol{W}|\mathcal{D}, \boldsymbol{\mu}_p, \boldsymbol{\sigma}_p^2)) \geq 0$, the ELBO lower bounds log marginal likelihood $\log p(\mathcal{D}|\boldsymbol{\mu}_p, \boldsymbol{\sigma}_p^2) \geq \mathcal{L}(\boldsymbol{\mu}_q, \boldsymbol{\sigma}_q^2; \boldsymbol{\mu}_p, \boldsymbol{\sigma}_p^2)$. Previous work [7, 38] introduces low variance updates for mean-field VI by reparametrizing weights $\boldsymbol{W}$. Bayesian predictions are approximated by integrating the learned variational posterior $q^*(\boldsymbol{W}|\boldsymbol{\mu}_q, \boldsymbol{\sigma}_q^2)$ as $p(\boldsymbol{y}^*|\boldsymbol{x}^*, \mathcal{D}) \approx \mathbb{E}_{q^*(\boldsymbol{W}|\boldsymbol{\mu}_q, \boldsymbol{\sigma}_q^2)} p(\boldsymbol{y}^*|f_{\boldsymbol{W}}(\boldsymbol{x}^*))$.

For BNNs, prior means are most often set to zero $\boldsymbol{\mu}_p = \boldsymbol{0}$, and prior variances are set to a constant $\boldsymbol{\sigma}_p^2 = \gamma \boldsymbol{1}$ (often scaled by the size of a previous layer [55, 13]). We consider a more general setup where the weights of a BNN are defined by a hierarchical model $p(\boldsymbol{W}|\boldsymbol{\mu}_p, \boldsymbol{\sigma}_p^2) p(\boldsymbol{\mu}_p) p(\boldsymbol{\sigma}^2)$ and aim to perform inference over weights $\boldsymbol{W}$ and $\boldsymbol{\mu}_p, \boldsymbol{\sigma}_p^2$. For clarity, we will refer to $\boldsymbol{\mu}_p, \boldsymbol{\sigma}_p^2$ as *prior parameters*, leaving the term *hyperparameters* for any other free variables.

### 2.1 Deriving collapsed variational bounds

In this section, we develop an efficient optimization scheme for variational BNNs with weights defined in a hierarchical fashion, where the inference over prior parameters is done analytically and prior parameters are subsequently marginalized out. This defines new learning objectives for learning variational BNNs, which we show by providing specific examples later in the text. More specifically, we apply collapsed variational bounds [36, 43, 47, 26] to BNNs, and present a general, systematic way of deriving novel, tighter lower bounds on the model evidence which are useful for learning variational posteriors over weights. The idea behind a collapsed bound is simple: Suppose we maximize $f(x, y)$ and we can derive the optimal $x^*(y) = \operatorname{argmax}_x f(x, y)$. Substituting $x^*$ into $f$ results in $f(x^*(y), y)$, which only depends on $y$, is easier to optimize, and has the property that $\forall_y f(x^*(y), y) \geq f(x, y)$. Collapsed variational bounds apply this reasoning to the ELBO with factorized variational posteriors taking the roles of $x$ and $y$.

**Inferring prior parameters in BNN.** We consider inference over both network weights $\boldsymbol{W}$ and prior parameters $\boldsymbol{\mu}_p$ and $\boldsymbol{\sigma}_p^2$, which we treat as latent variables and subsequently discuss collapsing the inference w.r.t. prior parameters $\boldsymbol{\mu}_p$ and $\boldsymbol{\sigma}_p^2$. We use the factorization $q(\boldsymbol{W}, \boldsymbol{\mu}_p, \boldsymbol{\sigma}_p^2|\boldsymbol{\mu}_q, \boldsymbol{\sigma}_q^2) = q(\boldsymbol{W}|\boldsymbol{\mu}_q, \boldsymbol{\sigma}_q^2) q(\boldsymbol{\mu}_p, \boldsymbol{\sigma}_p^2)$, where we omit the variational parameters of $q(\boldsymbol{\mu}_p, \boldsymbol{\sigma}_p^2)$ for concise notation. Learning $q(\boldsymbol{\mu}_p, \boldsymbol{\sigma}_p^2)$ under this assumption does not complicate computing the BNN's predictive distribution $p(\boldsymbol{y}|\boldsymbol{x}, \mathcal{D})$, i.e. $\boldsymbol{\mu}_q, \boldsymbol{\sigma}_q^2$ do not need to be sampled at the prediction time, since,

$$p(\boldsymbol{y}|\boldsymbol{x}, \mathcal{D}) = \mathbb{E}_{p(\boldsymbol{W}, \boldsymbol{\mu}_p, \boldsymbol{\sigma}_p^2|\mathcal{D})} p(\boldsymbol{y}|f_{\boldsymbol{W}}(\boldsymbol{x})) \approx \mathbb{E}_{q(\boldsymbol{W}, \boldsymbol{\mu}_p, \boldsymbol{\sigma}_p^2)} p(\boldsymbol{y}|f_{\boldsymbol{W}}(\boldsymbol{x})) = \mathbb{E}_{q(\boldsymbol{W}|\boldsymbol{\mu}_q, \boldsymbol{\sigma}_q^2)} p(\boldsymbol{y}|f_{\boldsymbol{W}}(\boldsymbol{x})). \tag{2}$$

To apply inference, we invoke the variational principle which lower bounds the log marginal likelihood $\log p(\mathcal{D})$ as:

$$\log p(\mathcal{D}) \geq \mathbb{E}_{q(\boldsymbol{W}|\boldsymbol{\mu}_q, \boldsymbol{\sigma}_q^2) q(\boldsymbol{\mu}_p, \boldsymbol{\sigma}_p^2)} \log \frac{p(\mathcal{D}, \boldsymbol{W}, \boldsymbol{\mu}_p, \boldsymbol{\sigma}_p^2)}{q(\boldsymbol{W}|\boldsymbol{\mu}_q, \boldsymbol{\sigma}_q^2) q(\boldsymbol{\mu}_p, \boldsymbol{\sigma}_p^2)} = \mathcal{L}^q(\boldsymbol{\mu}_q, \boldsymbol{\sigma}_q^2, q(\boldsymbol{\mu}_p, \boldsymbol{\sigma}_p^2)). \tag{3}$$

The bound in Eq. (3) reduces to $\mathcal{L}(\boldsymbol{\mu}_q, \boldsymbol{\sigma}_q^2)$ in Eq. (1) when $q(\boldsymbol{\mu}_p, \boldsymbol{\sigma}_p^2) = q(\boldsymbol{\mu}_p)q(\boldsymbol{\sigma}_p^2)$ and $q(\boldsymbol{\mu}_p), q(\boldsymbol{\sigma}_p^2)$ are delta functions. If we are able to analytically derive the distribution $q^*(\boldsymbol{\mu}_p, \boldsymbol{\sigma}_p^2)$ maximizing

$\mathcal{L}^q$, we can substitute the resulting distribution into Eq. (3) to derive a tighter (collapsed) bound on the marginal likelihood, saving the need to perform coordinate ascent or gradient updates to learn the prior parameters:

$$\log p(\mathcal{D}) \geq \mathbb{E}_{q(\boldsymbol{W}|\boldsymbol{\mu}_q,\boldsymbol{\sigma}_q^2)q^*(\boldsymbol{\mu}_p,\boldsymbol{\sigma}_p^2)} \log \frac{p(\mathcal{D},\boldsymbol{W},\boldsymbol{\mu}_p,\boldsymbol{\sigma}_p^2)}{q(\boldsymbol{W}|\boldsymbol{\mu}_q,\boldsymbol{\sigma}_q^2)q^*(\boldsymbol{\mu}_p,\boldsymbol{\sigma}_p^2)} = \mathcal{L}^*(\boldsymbol{\mu}_q,\boldsymbol{\sigma}_q^2), \qquad (4)$$

where $\mathcal{L}^*(\boldsymbol{\mu}_q,\boldsymbol{\sigma}_q^2) \geq \mathcal{L}^q(\boldsymbol{\mu}_q,\boldsymbol{\sigma}_q^2,q(\boldsymbol{\mu}_p,\boldsymbol{\sigma}_p^2))$ and $\mathcal{L}^*$ depends only on the variational parameters $\boldsymbol{\mu}_q, \boldsymbol{\sigma}_q^2$. Collapsed variational bounds are desirable as they have been shown to make learning significantly more efficient [64]. The tighter bound in Eq. (4) has been also referred to as a KL corrected bound [36] and marginal VI bound [43, 47].

**Collapsing the variational bound for BNNs.** We now provide a method to derive a collapsed variational bound by maximizing $\mathcal{L}^q(\boldsymbol{\mu}_q,\boldsymbol{\sigma}_q^2,q(\boldsymbol{\mu}_p,\boldsymbol{\sigma}_p^2))$ w.r.t. the posterior over weights prior parameters $q(\boldsymbol{\mu}_p,\boldsymbol{\sigma}_p^2)$ in BNNs. We use the property that the BNN likelihood term $\log p(\boldsymbol{y}|\boldsymbol{x}, f_{\boldsymbol{W}}(\boldsymbol{x}))$ does not depend on the prior parameters $\boldsymbol{\mu}_p$ and $\boldsymbol{\sigma}_p^2$, i.e. $p(\mathcal{D}|\boldsymbol{W},\boldsymbol{\mu}_p,\boldsymbol{\sigma}_p^2) = p(\mathcal{D}|\boldsymbol{W})$. The bound $\mathcal{L}^q$ from Eq. (3) decomposes into three terms:

$$\mathcal{L}^q\big(\boldsymbol{\mu}_q,\boldsymbol{\sigma}_q^2,q(\boldsymbol{\mu}_p,\boldsymbol{\sigma}_p^2)\big) = \mathcal{L}_{data}(\boldsymbol{\mu}_q,\boldsymbol{\sigma}_q^2) + \Phi(\boldsymbol{\mu}_q,\boldsymbol{\sigma}_q^2,q(\boldsymbol{\mu}_p,\boldsymbol{\sigma}_p^2)) + \mathcal{H}[q(\boldsymbol{W}|\boldsymbol{\mu}_q,\boldsymbol{\sigma}_q^2)], \qquad (5)$$

where $\mathcal{H}[q(\boldsymbol{W}|\boldsymbol{\mu}_q,\boldsymbol{\sigma}_q^2)]$ denotes the differential entropy, and we have defined

$$\Phi\big(\boldsymbol{\mu}_q,\boldsymbol{\sigma}_q^2,q(\boldsymbol{\mu}_p,\boldsymbol{\sigma}_p^2)\big) = \mathbb{E}_{q(\boldsymbol{\mu}_p,\boldsymbol{\sigma}_p^2)}\left[\mathbb{E}_{q(\boldsymbol{W}|\boldsymbol{\mu}_q,\boldsymbol{\sigma}_q^2)} \log p(\boldsymbol{W}|\boldsymbol{\mu}_p,\boldsymbol{\sigma}_p^2)\right] - D_{KL}\big(q(\boldsymbol{\mu}_p,\boldsymbol{\sigma}_p^2)||p(\boldsymbol{\mu}_p)p(\boldsymbol{\sigma}_p^2)\big).$$
$$(6)$$

Only $\Phi$ depends on the posterior $q(\boldsymbol{\mu}_p,\boldsymbol{\sigma}_p^2)$, and therefore finding $q^*(\boldsymbol{\mu}_p,\boldsymbol{\sigma}_p^2)$ can be done by maximizing $\Phi(\boldsymbol{\mu}_q,\boldsymbol{\sigma}_q^2,q(\boldsymbol{\mu}_p,\boldsymbol{\sigma}_p^2))$.

We can significantly simplify this by noting that the objective in Eq. (6) is analogous to Eq. (1), hence optimizing $\Phi$ takes the form of a nested VI problem: we want to infer $q(\boldsymbol{\mu}_p,\boldsymbol{\sigma}_p^2)$ where the "data" distribution is replaced by the current variational posterior over weights $q(\boldsymbol{W}|\boldsymbol{\mu}_q,\boldsymbol{\sigma}_q^2)$. This implies the solutions $q^*(\boldsymbol{\mu}_p,\boldsymbol{\sigma}_p^2)$ are straightforward to derive in closed-form for priors/approximate posteriors from the exponential family [5, 6]. Specifically, we can write the maximizer $q^*(\boldsymbol{\mu}_p,\boldsymbol{\sigma}_p^2)$ as:

$$\log q^*(\boldsymbol{\mu}_p,\boldsymbol{\sigma}_p^2) \propto \log p(\boldsymbol{\mu}_p) + \log p(\boldsymbol{\sigma}_p^2) + \mathbb{E}_{q(\boldsymbol{W}|\boldsymbol{\mu}_q,\boldsymbol{\sigma}_q^2)} \log p(\boldsymbol{W}|\boldsymbol{\mu}_p,\boldsymbol{\sigma}_p^2). \qquad (7)$$

Substituting the optimal variational posterior $q^*(\boldsymbol{\mu}_p,\boldsymbol{\sigma}_p^2)$ into Eq. (6) gives us $\Phi^*$, and we can substitute this into Eq. (5) to give our final collapsed bound $\mathcal{L}^*(\boldsymbol{\mu}_q,\boldsymbol{\sigma}_q^2) = \mathcal{L}_{data} + \Phi^* + \mathcal{H}[q(\boldsymbol{W}|\boldsymbol{\mu}_p,\boldsymbol{\sigma}_p^2)]$. Furthermore, when calculating $\Phi^*$, the inferred prior parameters $q^*(\boldsymbol{\mu}_p,\boldsymbol{\sigma}_p^2)$ can be analytically integrated out in many cases, i.e. the outer expectation on the RHS in Eq. (6) can be solved in closed-form (we provide examples in Section 2.2), leading to a concise optimization objective.

The final optimization target to learn the variational posterior $q(\boldsymbol{W}|\boldsymbol{\mu}_q,\boldsymbol{\sigma}_q^2)$ is given by $\mathcal{L}^*(\boldsymbol{\mu}_q,\boldsymbol{\sigma}_q^2)$. Compared to gradient learning of $\boldsymbol{\mu}_p$ and $\boldsymbol{\sigma}_p^2$, eliminating $\boldsymbol{\mu}_p$ and $\boldsymbol{\sigma}_p^2$ from $\mathcal{L}^*$ performs better (as we show in Section 4), and halves both the memory requirement and time taken per update. It is important to remember that optimizing $\mathcal{L}^*$ learns both the variational parameters $\boldsymbol{\mu}_q, \boldsymbol{\sigma}_q^2$ and the posterior over prior parameters $q^*(\boldsymbol{\mu}_p,\boldsymbol{\sigma}_p^2)$, but the latter is implicit as we express it as a function of $\boldsymbol{\mu}_q, \boldsymbol{\sigma}_q^2$ given by Eq. (7).

In summary, there are four steps to derive a collapsed bound $\mathcal{L}^*$, which we directly use to learn the variational posterior of a BNN. These steps are given in Algorithm 1. We next give two concrete examples of collapsed bounds.

## 2.2 Examples of tighter ELBOs

**Learn prior means, fix prior variances.** As an easier example, we first consider fixing the prior variance of weights, and learning prior means. We follow the four steps of Algorithm 1 to derive our collapsed bound. Detailed derivations are in Appendix B.

*Step I:* This step defines a model and the family of variational approximations. We choose $p(\boldsymbol{W}|\boldsymbol{\mu}_p) = \mathcal{N}(\boldsymbol{W}|\boldsymbol{\mu}_p,\gamma\mathbf{1})$ with a Gaussian prior $p(\boldsymbol{\mu}_p|\alpha) = \mathcal{N}(\boldsymbol{\mu}_p|\mathbf{0},\alpha\mathbf{1})$, where $\alpha$ and $\gamma$ are

---

**Algorithm 1** Deriving a collapsed bound $\mathcal{L}^*$ for BNN in four steps.

**Step I:**   Choose priors $p(\boldsymbol{\mu}_p), p(\boldsymbol{\sigma}_p^2), p(\boldsymbol{W}|\boldsymbol{\mu}_p, \boldsymbol{\sigma}_p^2)$,
and approximate posteriors $q(\boldsymbol{\mu}_p, \boldsymbol{\sigma}_p^2), q(\boldsymbol{W}|\boldsymbol{\mu}_q, \boldsymbol{\sigma}_q^2)$ over network weights.

**Step II:**   Calculate optimal prior parameters $q^*(\boldsymbol{\mu}_p, \boldsymbol{\sigma}_p^2)$ using Eq. (7).

**Step III:**   Form $\Phi^*$ and solve $\mathbb{E}_{q^*(\boldsymbol{\mu}_p, \boldsymbol{\sigma}_p^2)}\left[\mathbb{E}_{q(\boldsymbol{W}|\boldsymbol{\mu}_q, \boldsymbol{\sigma}_q^2)} \log p(\boldsymbol{W}|\boldsymbol{\mu}_p, \boldsymbol{\sigma}_p^2)\right]$ in Eq. (6).

**Step IV:**   The collapsed bound is $\mathcal{L}^* = \mathcal{L}_{data} + \Phi^* + \mathcal{H}[q(\boldsymbol{W}|\boldsymbol{\mu}_q, \boldsymbol{\sigma}_q^2)]$.

---

hyperparameters. We also employ a mean-field Gaussian posterior $q(\boldsymbol{W}|\boldsymbol{\mu}_q, \boldsymbol{\sigma}_q^2)$ and a Gaussian posterior $q(\boldsymbol{\mu}_p)$.

*Step II:* This step analytically finds the optimal variational distribution $q^*(\boldsymbol{\mu}_p)$. Substituting our distributions into Eq. (7) for every coordinate $\mu_p$ gives us

$$\log q^*(\mu_p) \propto -\frac{(\mu_q - \mu_p)^2}{2\gamma} - \frac{\mu_p^2}{2\alpha}. \tag{8}$$

In this case the variational posterior $q^*(\boldsymbol{\mu}_p)$ matches the true posterior, as the above dependency is equivalent Bayesian inference over $\boldsymbol{\mu}_p$ given Gaussian likelihood/prior and observation $\boldsymbol{\mu}_q$, so $q^*(\boldsymbol{\mu}_p) = \mathcal{N}(\boldsymbol{\mu}_p|\frac{\alpha}{\alpha+\gamma}\boldsymbol{\mu}_q, \frac{\alpha\gamma}{\alpha+\gamma}\mathbf{1})$.

*Step III:* This step forms $\Phi^*$ by calculating the divergences between optimal posteriors $q^*$ and priors and simplifies the derived optimization objective by marginalizing out approximate prior parameters. The divergence between two Gaussians $D_{KL}(q^*(\boldsymbol{\mu}_p)||p(\boldsymbol{\mu}_p))$ is straightforward to compute. Next we solve the integral $\mathcal{I} = \mathbb{E}_{q^*(\boldsymbol{\mu}_p)}[\mathbb{E}_{q(\boldsymbol{W}|\boldsymbol{\mu}_q, \boldsymbol{\sigma}_q^2)} \log p(\boldsymbol{W}|\boldsymbol{\mu}_p, \boldsymbol{\sigma}_p^2)]$:

$$\mathcal{I} \triangleq - \mathop{\mathbb{E}}_{\mathcal{N}(\boldsymbol{\mu}_p|\frac{\alpha}{\alpha+\gamma}\boldsymbol{\mu}_q, \frac{\alpha\gamma}{\alpha+\gamma}\mathbf{1})}\left[\frac{\mathbf{1}^T\boldsymbol{\sigma}_q^2 + (\boldsymbol{\mu}_q - \boldsymbol{\mu}_p)^T(\boldsymbol{\mu}_q - \boldsymbol{\mu}_p)}{2\gamma}\right] \triangleq -\frac{\mathbf{1}^T\boldsymbol{\sigma}_q^2}{2\gamma} - \frac{\gamma\boldsymbol{\mu}_q^T\boldsymbol{\mu}_q}{2(\alpha+\gamma)^2}.$$

*Step IV:* This step forms a collapsed bound that will be used to learn variational posteriors. Substituting the above into $\mathcal{L}^* = \mathcal{L}_{data} + \Phi^* + \mathcal{H}[q(\boldsymbol{W}|\boldsymbol{\mu}_p, \boldsymbol{\sigma}_p^2)]$ results in our new bound:

$$\mathcal{L}_m^*(\boldsymbol{\mu}_q, \boldsymbol{\sigma}_q^2) \triangleq \mathcal{L}_{data}(\boldsymbol{\mu}_q, \boldsymbol{\sigma}_q^2) - \frac{1}{2\gamma}\left[\mathbf{1}^T\boldsymbol{\sigma}_q^2 + \alpha_{reg}\boldsymbol{\mu}_q^T\boldsymbol{\mu}_q\right] + \frac{1}{2}\mathbf{1}^T \log\boldsymbol{\sigma}_q^2 + \frac{D}{2}\log\alpha_{reg}, \tag{9}$$

where we have defined the hyperparameter $\alpha_{reg} = \gamma/(\gamma + \alpha) \in (0, 1)$, and for fixed $\gamma$ there is a 1-to-1 relation between $\alpha$ and $\alpha_{reg}$.

When $\alpha \to 0$ (so $\alpha_{reg} \to 1$) we recover the standard expression for the ELBO given by Eq. (1) with $q(\boldsymbol{W}|\boldsymbol{\mu}_q, \boldsymbol{\sigma}_q^2)$ and prior $\mathcal{N}(\boldsymbol{W}|\mathbf{0}, \gamma\mathbf{1})$. This corresponds to a dogmatic prior $p(\boldsymbol{\mu}_p) = \mathcal{N}(\boldsymbol{\mu}_p|\mathbf{0}, \alpha\mathbf{1})$ converging to a delta spike at 0, and no inference of $\boldsymbol{\mu}_p$. When we allow for inference, the regularization of the (approximate) posterior mean $\boldsymbol{\mu}_q$ decreases. When $\alpha_{reg} \to 0$ (uninformative prior) the modeling can be prone to over-fitting due to insufficient regularization in the model specification. Note that this is reflected by the term $\frac{D}{2}\log\alpha_{reg}$ which can be interpreted as an Occam's Razor penalty [59]: this term penalizes small $\alpha_{reg}$ as it diverges to negative infinity when $\alpha_{reg} \to 0$. Since $\alpha_{reg} \in (0, 1)$, this enables finding well-performing values of $\alpha_{reg}$ by evaluating $\mathcal{L}_m^*$ (without having a validation set) [30], and we show this in our experiments in Section 4.

We find that $\alpha_{reg} \in (0.01, 0.1)$ significantly improves upon the predictions of MF-VI with fixed Gaussian priors. For existing implementations learning BNNs with Gaussian variational posterior/prior we suggest the default setting $\alpha_{reg} = 0.05$ in front of the mean regularization in the expression for KL divergence to instantaneously improve the predictions (as opposed to down-weighting the whole KL divergence term). Larger $\alpha_{reg}$ increases the strength of the regularization of the model.

**Learn both prior means and variances.** We now discuss a scheme to learn hyperparameters of both prior means and variances. We follow the same four steps from Algorithm 1. We again defer the detailed derivations to Appendix B.

*Step I:* $p(\boldsymbol{W}|\boldsymbol{\mu}_p, \boldsymbol{\tau}_p) = \mathcal{N}(\boldsymbol{W}|\boldsymbol{\mu}_p, \frac{1}{\boldsymbol{\tau}_p})$ and $p(\boldsymbol{\mu}_p|t\boldsymbol{\tau}) = \mathcal{N}(\boldsymbol{\mu}_p|\mathbf{0}, \frac{1}{t\boldsymbol{\tau}})$, $p(\boldsymbol{\tau}) = \mathcal{G}(\boldsymbol{\tau}|\boldsymbol{\alpha}, \boldsymbol{\beta})$. We again consider posteriors $q(\boldsymbol{W}|\boldsymbol{\mu}_q, \boldsymbol{\sigma}_q^2) = \mathcal{N}(\boldsymbol{W}|\boldsymbol{\mu}_q, \boldsymbol{\sigma}_q^2)$, $q(\boldsymbol{\mu}_p|\boldsymbol{\tau}_p) = \mathcal{N}(\boldsymbol{\mu}_p)$ and $q(\boldsymbol{\tau}_p) = \mathcal{G}(\boldsymbol{\tau}_p)$.

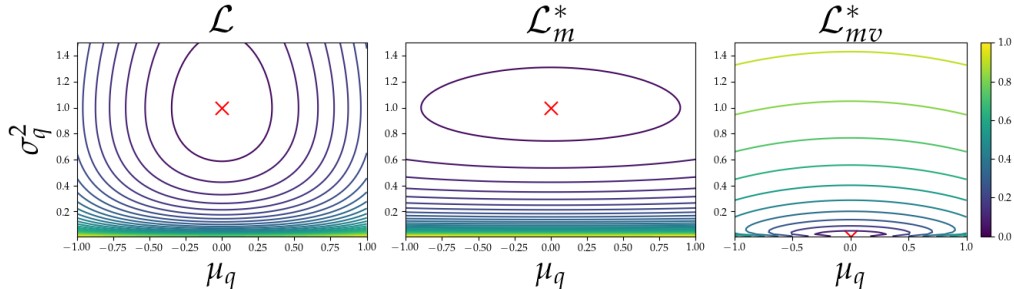

Figure 1: Comparison of regularizing terms in $\mathcal{L}$, $\mathcal{L}_m^*$ and $\mathcal{L}_{mv}^*$ scaled to unity. In the absence of gradients from $\mathcal{L}_{data}$ (approximate) posteriors over weights converge to the blue region (red cross-mark). Optimizing $\mathcal{L}$ prunes weights to prior Gaussians which still influence predictions causing under-fitting (see Section 4) and is opposite to optimizing $\mathcal{L}_{mv}^*$ which prunes to spiked Gaussians.

*Step II*: The optimal posteriors $q^*$ are given by $q^*(\boldsymbol{\mu}_p) = \mathcal{N}(\boldsymbol{\mu}_p | \frac{1}{1+t}\boldsymbol{\mu}_q, \frac{1}{(1+t)\boldsymbol{\tau}_p})$ and $q^*(\boldsymbol{\tau}_p) = \mathcal{G}(\boldsymbol{\tau}_p | (\alpha + \frac{1}{2})\mathbf{1}, \beta + \frac{t}{2(1+t)}\boldsymbol{\mu}_q^2 + \frac{1}{2}\boldsymbol{\sigma}_q^2)$.

*Steps III and IV*: The bound $\mathcal{L}^*(\boldsymbol{\mu}_q, \boldsymbol{\sigma}_q^2)$ becomes,

$$\mathcal{L}_{mv}^*(\boldsymbol{\mu}_q, \boldsymbol{\sigma}_q^2) \triangleq \mathcal{L}_{data}(\boldsymbol{\mu}_q, \boldsymbol{\sigma}_q^2) - (\alpha + \frac{1}{2})\mathbf{1}^T \log\left[\beta\mathbf{1} + \frac{\delta}{2}\boldsymbol{\mu}_q^2 + \frac{1}{2}\boldsymbol{\sigma}_q^2\right] + \frac{1}{2}\mathbf{1}^T \log\boldsymbol{\sigma}_q^2, \quad (10)$$

where $\delta = t/(1+t)$. Eq. (10) recovers the case of setting prior mean $\boldsymbol{\mu}_p = \mathbf{0}$ and learning only prior variances $\boldsymbol{\sigma}_p^2$ when the prior precision $t\boldsymbol{\tau}_p$ over $\boldsymbol{\mu}_p$ goes to $\infty$, i.e. $t \to \infty$. Setting $\delta < 1$ in Eq. (10) weakens the regularization of posterior mean $\boldsymbol{\mu}_q$, allowing it to vary more. The regularizer in Eq. (10) is a decreasing function of $\mu_q$, i.e. without gradients from $\mathcal{L}_{data}$, the posterior mean $\mu_q$ of a weight $w$ converges to 0. In Section 4 we show that $\mathcal{L}_{mv}^*$ outperforms standard MF-VI.

**Comparison of regularizing terms.** We compare the regularization terms on $\boldsymbol{\mu}_q$ and $\boldsymbol{\sigma}_q^2$ in the standard mean-field ELBO in Eq. (1) and introduced ELBOs in Eq. (9) and Eq. (10) in Figure 1, where we use prior the $p(w) = \mathcal{N}(w|0,1)$. We consider what happens in the absence of gradients from the data term $\mathcal{L}_{data}$ to better understand the bounds. In Figure 1 (left) we see that Gaussian posteriors (in the absence of data) optimized with Eq. (1) converge to their prior $\mathcal{N}(w|0,1)$ [7, 68]. In Figure 1 (middle) we are down-weighting $\boldsymbol{\mu}_q^T\boldsymbol{\mu}_q$ in Eq. (9), hence weakening the regularization of $\boldsymbol{\mu}_q$. This allows posterior means to vary more, and can be interpreted as scaling the $\mu_p$ axis in Figure 1 by $\sqrt{\alpha}$. As we show in Section 4, this enables posterior means $\mu_q$ to saturate activations and reduce the noise from pruned weights. Optimizing Eq. (10) is roughly opposite to Eq. (1) and causes weights to be pruned to $\mathcal{N}(0, 2\beta)$ (Figure 1 right, where $\beta = 0.01$). Weights pruned in this way do influence the predictions for small $\beta$ and this fixes the excessive injection of noise by pruned weights, as we show in Section 4.

# 3   Related work

Performance of BNNs is known to be sensitive to prior hyperparameters [54, 71, 31], but optimizing hyperparameters in BNNs has not been widely adopted, although there are exceptions [50, 72]. Some authors heuristically propose using a Gaussian centered on the MAP estimate as a prior for BNNs [8, 39]. The Laplace approximation has been used for hyperparameter selection in BNNs [49, 31, 61], but it does not use the ELBO. Hierarchical models in BNNs have been explored in many works [23, 22, 46, 33, 34, 58, 3, 12, 2]. The closest to our approach is learning a hierarchical horseshoe prior over BNNs weights [23]. However, their approach is computationally costly, does not consider collapsed bounds to derive tighter ELBOs, and does not learn prior means. There is an active discussion in the community as to whether the true BNN posterior provides satisfactory predictions with work supporting the Bayesian approach [54, 74, 32, 1] and arguing against [70]. Similarly, there are conflicting views on the performance of MF-VI in BNNs with fixed Gaussian priors as some authors claim it can work well [17] and others arguing against [63, 19, 18]. A number of problems of mean-field VI BNNs have been exposed in shallow networks including over-pruning [68] and poor uncertainty in output space [19, 18, 23]. To resolve under-fitting of mean-field BNNs, previous work has re-weighted terms arising from the KL divergence e.g. [45].

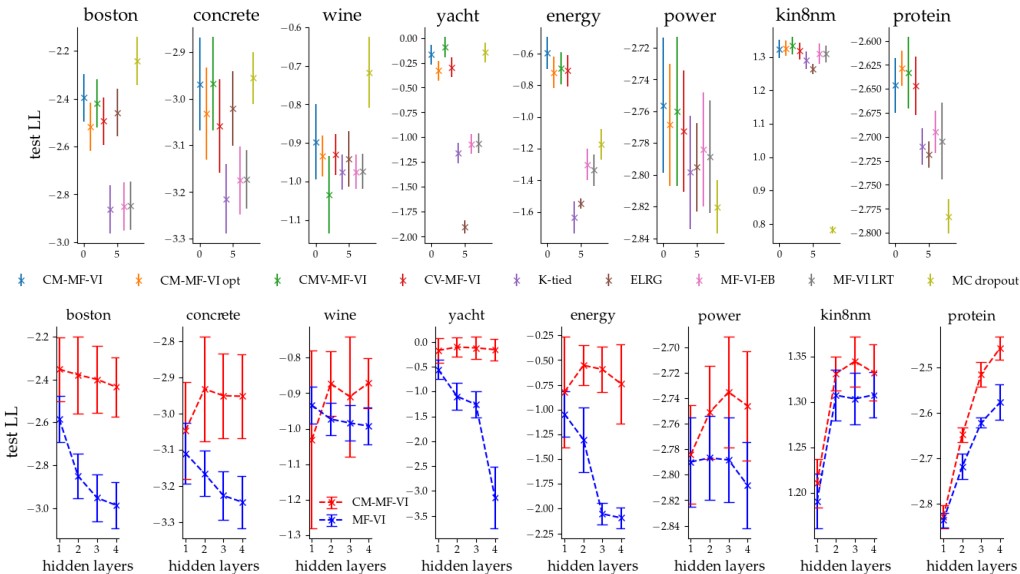

Figure 2: Top: Comparison of collapsed bounds with other algorithms on UCI regression data sets. Bottom: CM-MF-VI (red) and standard MF-VI with a default isotropic Gaussian prior (blue) for deeper networks, where CM-MF-VI avoids severe degradation of predictions.

## 4 Experiments

In this section we explore the predictive performance of the introduced variational bounds and benchmark them together with other algorithms. We find that using the bounds in Eq. (9) and Eq. (10) significantly improves the predictive performance of standard MF-VI BNNs (fixed Gaussian priors). We also find that learning the prior means according to Eq. (9) is more effective and gives sufficiently good predictive performance, whereas learning prior variances (Eq. (10)) mostly does not lead to significant gains in the quality of predictions. We defer the details of the experimental setup to Appendix C and report additional experimental results in Appendix D.

The algorithms we propose in this paper optimize various tighter ELBOs: CM-MF-VI optimizes $\mathcal{L}_m$ from Eq. (9); CM-MF-VI OPT additionally optimizes $\alpha_{reg}$ using the tighter ELBO (Eq. (9)); CV-MF-VI optimizes for just prior variances, using the bound $\mathcal{L}_{mv}$ from Eq. (10) when $\delta \to 1$ (prior mean $\boldsymbol{\mu}_p$ fixed to $\mathbf{0}$); and CMV-MF-VI optimizes both prior means and variances from Eq. (10). We compare to the following baselines: MF-VI LRT learns zero-mean mean-field Gaussian priors with fixed variances [55] (which we call standard MF-VI) using local reparametrization [38]; MF-VI FV learns a mean-field Gaussian posterior with fixed variances [68]; MF-VI BD is mean-field Gaussian VI while down-weighting KL penalty by $0.5$; MF-VI EB learns prior means with ELBO gradient updates [51, 72]; BBB is Bayes by Backprop using the ADAM optimizer (as opposed to SGD in [7]); MF-VI $k$-tied normal distribution [63]; ELRG/SLANG learn low rank posteriors [66, 53].

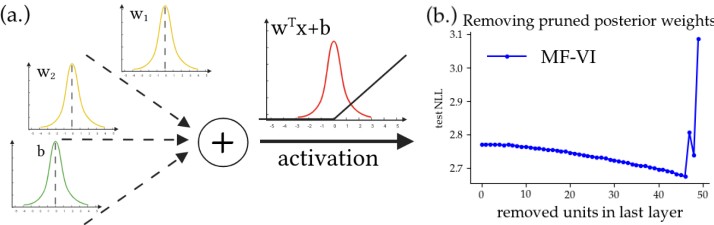

Figure 3: (a.) Pruned weights in standard MF-VI contribute noise to predictions as noise leaks through the activation function. (b.) Removing pruned weights from BNN can *improve* performance, as the removed weights contribute only to the predictive variance and cause under-fitting.

| | 400 units | | 800 units | |
|---|---|---|---|---|
| algorithm | test NLL ↓ | test ER ↓ | test NLL ↓ | test ER ↓ |
| CM-MF-VI | $0.047 \pm 0.006$ | $1.34 \pm 0.24\%$ | $0.048 \pm 0.005$ | $1.42 \pm 0.26\%$ |
| CV-MF-VI | $0.068 \pm 0.007$ | $2.13 \pm 0.34\%$ | $0.066 \pm 0.003$ | $2.09 \pm 0.10\%$ |
| CMV-MF-VI | $0.049 \pm 0.005$ | $1.45 \pm 0.25\%$ | $0.052 \pm 0.004$ | $1.60 \pm 0.01\%$ |
| CM-MF-VI (4000 batch) | $\mathbf{0.041 \pm 0.002}$ | $\mathbf{1.31 \pm 0.04}\%$ | $\mathbf{0.042 \pm 0.002}$ | $\mathbf{1.20 \pm 0.06}\%$ |
| MF-VI LRT [38] | $0.094 \pm 0.002$ | $2.42 \pm 0.28\%$ | $0.099 \pm 0.001$ | $2.58 \pm 0.07\%$ |
| MF-VI BD | $0.092 \pm 0.002$ | $2.43 \pm 0.06\%$ | $0.099 \pm 0.004$ | $2.56 \pm 0.22\%$ |
| MF-VI FV | $0.052 \pm 0.001$ | $1.67 \pm 0.13\%$ | $0.053 \pm 0.004$ | $1.58 \pm 0.26\%$ |
| BBB ADAM [7] | $0.095 \pm 0.008$ | $2.48 \pm 0.45\%$ | $0.097 \pm 0.007$ | $2.49 \pm 0.33\%$ |
| ELRG-VI $K = 5$ [66] | $0.053 \pm 0.006$ | $1.54 \pm 0.18\%$ | $0.058 \pm 0.005$ | $1.68 \pm 0.17\%$ |
| K-TIED $K = 10$ [63] | $0.105 \pm 0.004$ | $2.67 \pm 0.16\%$ | $0.108 \pm 0.004$ | $2.61 \pm 0.17\%$ |
| MF-VI SGD [7] | $-$ | $1.82\%$ | $-$ | $1.99\%$ |
| SLANG K=32 [53] | $-$ | $1.72\%$ | $-$ | $-$ |

Table 1: Test NLL and error rate for vectorized MNIST classification with two hidden layer BNN. CM-MF-VI outperforms other algorithms and collapsed bounds improve upon standard MF-VI.

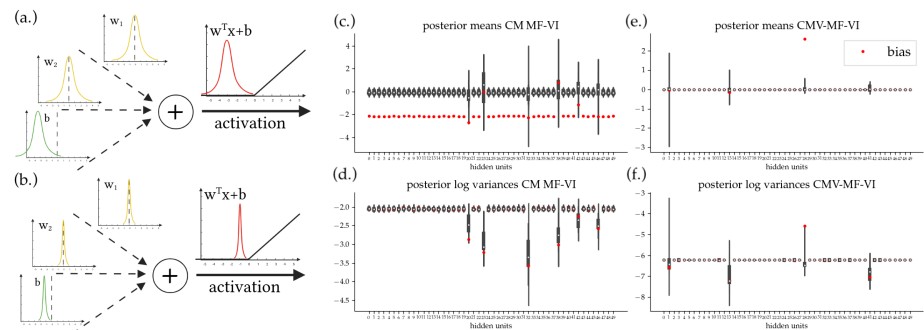

Figure 4: Learning prior parameters in BNN fixes under-fitting caused by over-pruning. In (c.),(d.),(e.) and (f.) we show the distribution of statistics of variational posterior over weights in the first layer. The top row plots (c.) and (e.) the posterior means, where red is the means of the bias parameters, and black is the means of the weight parameters into a hidden unit. The bottom row plots (d.) and (f.) show posterior log variances instead of posterior means. Most weights have been pruned. In CM-MF-VI, shown in (a.), units where posterior weights converge to prior have posterior biases set to low value, which saturates the activation stopping them from influencing the predictions. We see this in (c.), where the pruned units all have biases with negative posterior means. This saturation is made possible by weaker mean regularization in Eq. (9) governed by $\alpha_{reg}$. In CMV-MF-VI, shown in (b.), pruned units converge to spiked Gaussians. We see this in the (f.), where the posterior log variances of pruned units are all very small.

**UCI regression.** We first consider 20 train-test splits for 8 UCI regression data sets [14]. We learn 2 hidden layer BNNs (results for 1 hidden layer in Appendix D) with 50 units and ReLU activations [27, 21, 67], but we use a heteroscedastic observation model $p(\boldsymbol{y}|f_{\boldsymbol{W}}^1(\boldsymbol{x}), f_{\boldsymbol{W}}^2(\boldsymbol{x})) = \mathcal{N}(\boldsymbol{y}|f_{\boldsymbol{W}}^1(\boldsymbol{x}), \exp(f_{\boldsymbol{W}}^2(\boldsymbol{x})))$, where $f_{\boldsymbol{W}}^1(\boldsymbol{x}), f_{\boldsymbol{W}}^2(\boldsymbol{x})$ are two heads of the network. We optimize the objectives for 200K steps with the ADAM optimizer [37] with default settings. We report a comparison of test log-likelihood in Figure 2 (top). We observe that (i) the collapsed bounds CM-MF-VI, CV-MF-VI, CMV-MF-VI outperform MF-VI on all data (only CMV-MF-VI is worse on wine), (ii) bounds learning prior means CM-MF-VI, CMV-MF-VI outperform or match the bound learning only prior variances CV-MF-VI on all data sets, (iii) CM-MF-VI outperforms or matches MC-dropout on all data sets except for boston and wine (the performance of dropout is unstable unless $p$ is tuned per data set, e.g. it is poor on kin8nm and not shown). In addition, CM-MF-VI OPT matches the performance of CM-MF-VI, showing that the ELBO can be used to learn hyperparameter $\alpha_{reg}$. Next we report a comparison between CM-MF-VI and standard MF-VI as the depth of the network increases in Figure 2 (bottom). We see standard MF-VI starts to rapidly under-perform on most data sets (for larger data sets standard MF-VI performs well as it converges to MAP). Learning prior means/variances by optimizing bounds in Eq. (9) or Eq. (10) allows to mitigate rapid degradation of

predictions. We now explain why standard MF-VI performs badly, finding this is due to over-pruning in BNNs. We investigate this effect and find that learning prior means and variances mitigates the effect of pruned units on the predictive distribution.

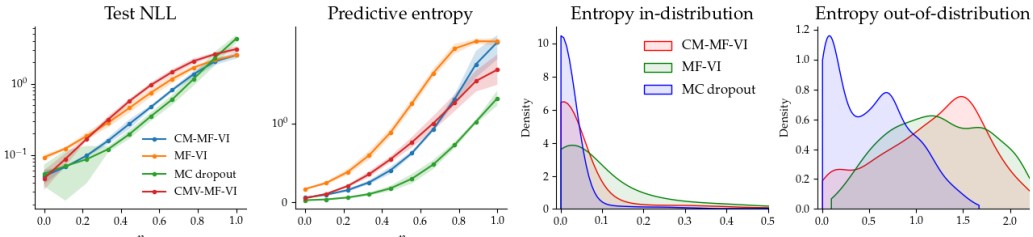

Figure 5: Comparison of standard MF-VI and collapsed bounds (CM-MF-VI, CV-MF-VI) for in-distribution and out-of-distribution data. Left: replacing $p\%$ of MNIST image with random bits. Right: learning BNN on MNIST (in-distribution) and testing on fashionMNIST (out-of-distribution).

**Over-pruning in BNNs.** Applying mean-field VI to BNN prunes most of the posterior weights, i.e. approximate posteriors converge to prior [7, 68, 23, 62]. While this is beneficial for compressing the model, pruned weights contribute noise to predictions and make modeling data difficult. With 1 hidden layer networks and the regular ELBO, output layer weights are set as close to 0 as possible [68], thereby reducing the noise from pruned weights in earlier layers. However, some noise still leaks through, as output layer weights are not quite delta-functions. This noise is one of the reasons for increasing under-confidence in output space as the size of the network increases [68, 16, 63, 66, 23, 45]. In Figure 3 we highlight this problem for 1 hidden layer MF-VI BNN with 50 hidden units, ReLU activations, zero-mean Gaussian prior over weights and heteroscedastic Gaussian likelihood learned with 200 random *boston* regression data points. In Figure 3 (right) we plot test NLL as pruned hidden layer's units are removed in ascending order of average KL penalties of incoming weights (only last 4 units are modeling data). Removing pruned weights lowers test NLL, meaning that pruned weights contribute only noise to predictions. This is schematically explained in Figure 3 (a), which shows pruned posterior weights injecting noise. See Appendix E for further description of over-pruning.

**Fixing under-fitting caused by over-pruning in BNNs.** We now explain why learning the hyperparameters of the Gaussian prior in the BNN mitigates problems arising from over-pruning. In short, learning the means and variances of Gaussian priors mitigates the influence of network units that have been pruned. These units otherwise cause under-fitting by contributing only to predictive variance. In Figure 4 we plot the distribution of learned mean-field Gaussian posterior means and log variances (middle and right) for units feeding into the output layer. We use a 1 hidden layer heteroscedastic BNN with 50 hidden units and ReLU activations learned on 200 data points from *boston* regression data set. We see that CM-MF-VI sets means of posterior biases to lower values ($\approx -2$) to saturate the activations for units where incoming weights converge to the prior. Saturated activations stop the pruned weights from contributing variance into the predictions and result in better modeling of data, as we schematically show in Figure 4 (a). For CMV-MF-VI, posterior weights corresponding to inactive units converge to delta spikes with very slightly negative means, in line with observations made in Figure 1, hence they do not influence predictions, shown in Figure 4 (b).

**MNIST classification with MLP.** Classifying vectorized MNIST images using two hidden layer network with ReLU activation is a standard benchmark for BNNs [7]. We demonstrate that collapsed bounds give large improvements in test NLL compared to using standard MF-VI and provides the best predictions across tested algorithms. We report test NLLs and test error rates (ER) averaged over 5 random seeds in Table 1. For 400 hidden units, CM-MF-VI/CMV-MF-VI achieve test NLL of 0.047, compared to 0.094 for standard MF-VI. CM-MF-VI (4000 batch) uses batches of 4000 images for an optimization step and achieves the best performance across the tested models, showing the developed algorithms can leverage low variance updates.

**Perturbed MNIST images.** We now investigate if the in-domain improvements of CM-MF-VI/CMV-MF-VI come at the cost of out-of-distribution (OOD) performance. We find this is not the case: OOD performance is as good as before. To show this, we learn a 2 hidden layer BNN with ReLU activations on vectorized MNIST images as previously, but test it on the fashionMNIST data

| test NLL ↓ / ER ↓ | MNIST | K-MNIST | F-MNIST | SVHN | CIFAR10 |
|---|---|---|---|---|---|
| CMV-MF-VI | **0.021 ± 0.001** | 0.152 ± 0.006 | **0.253 ± 0.006** | **0.313 ± 0.006** | 0.807 ± 0.005 |
| CM-MF-VI | **0.021 ± 0.001** | **0.141 ± 0.006** | 0.254 ± 0.005 | 0.315 ± 0.004 | 0.809 ± 0.009 |
| CV-MF-VI | 0.038 ± 0.003 | 0.239 ± 0.005 | 0.317 ± 0.006 | 0.321 ± 0.006 | 0.821 ± 0.005 |
| CM-MF-VI OPT | 0.024 ± 0.001 | 0.158 ± 0.004 | 0.258 ± 0.003 | **0.314 ± 0.004** | **0.789 ± 0.005** |
| MF-VI | 0.061 ± 0.001 | 0.319 ± 0.006 | 0.371 ± 0.003 | 0.340 ± 0.001 | 0.848 ± 0.009 |
| MAP | 0.048 ± 0.007 | 0.402 ± 0.027 | 0.336 ± 0.001 | 0.775 ± 0.012 | 1.134 ± 0.065 |
| MC dropout | 0.027 ± 0.001 | 0.222 ± 0.011 | 0.326 ± 0.007 | 0.400 ± 0.009 | 1.018 ± 0.017 |
| MF-VI EB | 0.060 ± 0.001 | 0.319 ± 0.003 | 0.372 ± 0.003 | 0.340 ± 0.005 | 0.843 ± 0.010 |
| CMV-MF-VI | 0.73 ± 0.02% | **4.00 ± 0.06%** | **8.90 ± 0.22%** | 8.49 ± 0.20% | 27.76 ± 0.37% |
| CM-MF-VI | **0.67 ± 0.04%** | **3.75 ± 0.30%** | 9.05 ± 0.31% | **8.21 ± 0.20%** | 27.63 ± 0.48% |
| CV-MF-VI | 1.24 ± 0.12% | 6.85 ± 0.10% | 11.47 ± 0.13% | 8.96 ± 0.20% | 28.39 ± 0.14% |
| CM-MF-VI OPT | 0.72 ± 0.05% | 4.23 ± 0.06% | 9.33 ± 0.07% | 8.47 ± 0.03% | **26.80 ± 0.41%** |
| MF-VI | 1.51 ± 0.05% | 9.17 ± 0.25% | 12.96 ± 0.28% | 9.20 ± 0.15% | 28.71 ± 0.16% |
| MAP | 1.16 ± 0.11% | 7.92 ± 0.36% | 11.94 ± 0.22% | 12.78 ± 0.12% | 34.63 ± 1.48% |
| MC dropout | 0.81 ± 0.05% | 6.07 ± 0.49% | 12.01 ± 0.17% | 9.78 ± 0.22% | 33.86 ± 0.42% |
| MF-VI EB | 1.53 ± 0.07% | 9.18 ± 0.19% | 12.96 ± 0.08% | 9.31 ± 0.28% | 28.56 ± 0.45% |

Table 2: Image classification with LeNet CNN. Collapsed bounds visibly outperform standard MF-VI.

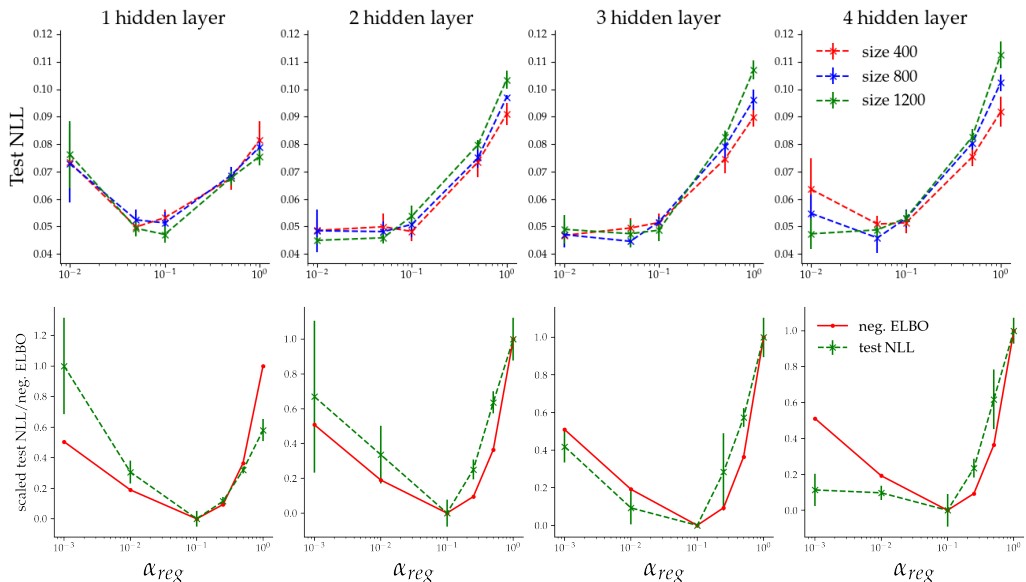

Figure 6: Top: performance of CM-MF-VI for different settings of hyperparameter $\alpha$ and different network architectures ($\alpha = 1$ is standard MF-VI). Setting $\alpha = 0.01$ visibly improves predictions compared to standard MF-VI (fixed Gaussian prior). Bottom: ELBO $\mathcal{L}_m^*$ correlates with test NLL.

set [73] and randomly perturbed MNIST images where every pixel is flipped with probability $p$. We report test NLL, predictive entropy for perturbed MNIST experiment in Figure 5 (left). We see that CM-MF-VI gives lower test NLL than MF-VI for any $p$, but it also maintains growing entropy when $p \rightarrow 1$ i.e. when we transition to OOD data. In Figure 5 (right) we plot histograms of predictive entropy for MNIST in-distribution data and OOD data fashionMNIST. CM-MF-VI significantly improves upon MF-VI's under-confidence without excessive reduction in uncertainty.

**Limitations of the algorithm: adjusting hyperparameters.** The bounds in Eq. (9) and Eq. (10) have hyperparameters, corresponding to the choice of hyper-prior. These additional parameters are the main limitation of the derived algorithms. For example, CM-MF-VI has one hyperparameter $\alpha_{reg}$ which allows us to control the strength of regularization of the model. We analyze the test predictive performance for different settings of $\alpha_{reg}$ for CM-MF-VI in Figure 6 (top) averaged over 5 random seeds. We found the setting $\alpha_{reg} = 0.075$ works well across many different problems and is robust to the choice of size/depth of the network. Increasing $\alpha_{reg}$ to 0.2 provides strong regularization and

| model | test NLL↓ / ER ↓ | CMV-MF-VI | CM-MF-VI | CV-MF-VI | MF-VI | MC dropout | MAP |
|---|---|---|---|---|---|---|---|
| RESNET18 | STL10 | **1.04 ± 0.00** | 1.10 ± 0.02 | 1.57 ± 0.01 | 1.61 ± 0.01 | 1.17 ± 0.05 | 1.64 ± 0.03 |
| | SVHN | 0.15 ± 0.00 | **0.14 ± 0.00** | 0.20 ± 0.01 | 0.22 ± 0.00 | 0.18 ± 0.00 | 0.35 ± 0.01 |
| | CIFAR100 | **1.43 ± 0.01** | 1.53 ± 0.00 | 2.00 ± 0.01 | 2.23 ± 0.03 | 1.75 ± 0.00 | 4.25 ± 0.05 |
| | CIFAR10 | 0.41 ± 0.00 | **0.39 ± 0.00** | 0.59 ± 0.00 | 0.68 ± 0.02 | 0.49 ± 0.00 | 0.93 ± 0.02 |
| SHUFFLENET | STL10 | **0.96 ± 0.01** | 0.99 ± 0.02 | 1.22 ± 0.05 | 1.70 ± 0.08 | 1.28 ± 0.01 | 1.78 ± 0.08 |
| | SVHN | **0.27 ± 0.01** | **0.26 ± 0.01** | 0.31 ± 0.01 | 0.31 ± 0.01 | 1.11 ± 0.01 | 0.32 ± 0.00 |
| | CIFAR100 | 2.03 ± 0.01 | **1.99 ± 0.02** | 2.21 ± 0.06 | 2.28 ± 0.02 | 3.08 ± 0.01 | 4.48 ± 0.03 |
| | CIFAR10 | **0.65 ± 0.00** | **0.65 ± 0.00** | 0.71 ± 0.01 | 0.72 ± 0.01 | 1.24 ± 0.00 | 1.10 ± 0.02 |
| ALEXNET | STL10 | 1.50 ± 0.07 | 1.48 ± 0.04 | 1.80 ± 0.07 | 1.86 ± 0.03 | **1.07 ± 0.06** | 1.80 ± 0.32 |
| | SVHN | 0.30 ± 0.00 | **0.28 ± 0.00** | 0.42 ± 0.01 | 0.51 ± 0.01 | 0.38 ± 0.01 | 0.72 ± 0.06 |
| | CIFAR100 | 2.24 ± 0.02 | **2.16 ± 0.06** | 2.62 ± 0.02 | 2.95 ± 0.04 | **2.19 ± 0.01** | 7.03 ± 0.14 |
| | CIFAR10 | 0.72 ± 0.01 | **0.69 ± 0.01** | 0.99 ± 0.01 | 1.19 ± 0.01 | 0.74 ± 0.01 | 1.79 ± 0.06 |
| RESNET18 | STL10 | 37.69 ± 0.25% | 39.75 ± 0.53% | 64.88 ± 0.54% | 66.58 ± 0.63% | 29.98 ± 1.17% | **29.30 ± 0.56**% |
| | SVHN | **3.76 ± 0.02**% | **3.75 ± 0.02**% | 5.26 ± 0.21% | 5.73 ± 0.09% | 4.11 ± 0.15% | 4.94 ± 0.07% |
| | CIFAR100 | **39.41 ± 0.39**% | 40.39 ± 0.37% | 53.78 ± 0.54% | 59.46 ± 0.72% | 45.51 ± 0.36% | 47.92 ± 0.34% |
| | CIFAR10 | 13.75 ± 0.06% | **13.34 ± 0.24**% | 20.22 ± 0.30% | 22.92 ± 1.14% | 16.36 ± 0.28% | 15.31 ± 0.35% |
| SHUFFLENET | STL10 | **34.48 ± 0.18**% | 34.79 ± 0.78% | 46.43 ± 2.83% | 73.74 ± 5.67% | 47.72 ± 1.12% | 41.41 ± 1.13% |
| | SVHN | 7.85 ± 0.32% | **7.31 ± 0.27**% | 8.90 ± 0.15% | 8.87 ± 0.21% | 26.19 ± 0.69% | 8.28 ± 0.19% |
| | CIFAR100 | 54.04 ± 0.07% | **52.66 ± 0.36**% | 58.95 ± 1.08% | 60.84 ± 0.54% | 73.96 ± 0.19% | 63.52 ± 0.22% |
| | CIFAR10 | 22.71 ± 0.61% | **22.68 ± 0.14**% | 24.66 ± 0.53% | 25.01 ± 0.47% | 42.54 ± 0.08% | 28.19 ± 0.59% |
| ALEXNET | STL10 | 56.59 ± 4.82% | 55.97 ± 2.99% | 74.38 ± 5.70% | 78.28 ± 1.32% | 37.33 ± 0.99% | **35.74 ± 0.67**% |
| | SVHN | 7.92 ± 0.20% | **7.41 ± 0.13**% | 11.61 ± 0.13% | 13.17 ± 0.20% | 8.87 ± 0.19% | 10.35 ± 0.71% |
| | CIFAR100 | 58.14 ± 0.83% | **54.91 ± 1.07**% | 66.82 ± 0.64% | 73.17 ± 1.39% | **55.39 ± 0.23**% | 60.14 ± 0.19% |
| | CIFAR10 | 24.35 ± 0.30% | **23.40 ± 0.20**% | 34.73 ± 0.34% | 41.63 ± 0.31% | 24.74 ± 0.43% | 26.04 ± 0.66% |

Table 3: Test NLL and error rates (ER) for the collapsed bounds on large scale CNN experiments. CMV-MF-VI, CM-MF-VI bounds provide visibly better predictions than MF-VI, which under-fits.

works well for data sets very prone to over-fitting. For models/data prone to under-fitting, setting $\alpha_{reg} \leq 0.025$ can give even better predictive performance. In Figure 6 (bottom) we show normalized values of ELBO to $\log p(\mathcal{D}, \alpha_{reg})$ with prior $\alpha_{reg} \sim Exp(5D)$, where $D$ is the number of network parameters, and test NLL for different values $\alpha_{reg}$. This has an important practical implication: approximately tuning $\alpha_{reg}$ can be guided by the ELBO. Figure 6 (bottom) can be compared with Fig 6 in [35] showing the same property using approximation of $\log p(\mathcal{D})$, as opposed to using ELBO.

**Image classification LeNet.** We now consider image classification with the LeNet architecture [44] on 6 data sets: MNIST, fashionMNIST, K-MNIST [10], CIFAR10, CIFAR100 [41] and SVHN [56]. We optimize the objectives for 800 epochs (except MAP for 50 epochs and MC dropout for 100 epochs as they tend to overfit) using batch size 512 and ADAM optimizer with default parameters. We report test NLL and test error rates (ER) averaged over 3 random seeds and standard deviation error bars in Table 2. We again observe the introduced bounds CM-MF-VI, CV-MF-VI and CMV-MF-VI outperform standard MF-VI in both test NLL and test error rate. CM-MF-VI performs slightly better than CMV-MF-VI, but the differences are not statistically significant. Both CM-MF-VI and CMV-MF-VI outperform MAP, CV-MF-VI and MC dropout in test NLL and test ER.

**Image classification large CNNs.** We follow by experimenting with larger CNNs: ResNet18 [25], ShuffleNet[48] and AlexNet[42]. We use CIFAR10, CIFAR100, STL10 [11] and SVHN. We again compare CMV-MF-VI, CV-MF-VI and CM-MF-VI to MF-VI, MC dropout and MAP. We optimize the objectives for 800 epochs (MAP and MC dropout early stopped at 200) with the default ADAM optimizer and the same data augmentation as in [57], and average results over 3 random seeds. We gather the results in Table 3. Experiments with large CNNs confirm our findings from previous experiments: (i) CM-MF-VI/CMV-MF-VI always outperform standard MF-VI by a visible margin and result in good predictive performance (e.g. outperforming SOTA VOGN [57]), (ii) learning prior means CM-MF-VI/CMV-MF-VI outperforms learning just prior variances (CV-MF-VI).

## 5 Conclusions

We developed a family of algorithms optimizing variational posteriors in BNNs based on collapsed variational bounds. We demonstrated that learning the prior parameters of BNN weights fixes their predictive under-confidence resulting in good empirical performance and robustness to over-fitting. The developed algorithms allowed us to demonstrate that the ELBO can be a suitable optimization target for learning hyperparameters of BNNs. Importantly, the introduced algorithms do not incur additional computational cost compared to applying MF-VI to BNNs and can be readily applied to improve the predictive performance of existing implementations. We hope that our approach will enable the practical use of VI based approximate inference in large network architectures.

## Acknowledgements

Marcin B. Tomczak is funded by a Cambridge Trust scholarship. Siddharth Swaroop is supported by an EPSRC DTP studentship and a Microsoft Research EMEA PhD Award. Andrew Y. K. Foong is supported by a Trinity Hall Research Studentship and the George and Lilian Schiff Foundation. Richard E. Turner is supported by Google, Amazon, ARM, Improbable, EPSRC grants EP/M0269571 and EP/L000776/1, and the UKRI Centre for Doctoral Training in the Application of Artificial Intelligence to the study of Environmental Risks (AI4ER). The authors thank David Burt for insightful discussions.

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
