## A  Different views on fixed prior variances

Suppose we aim to update variational posterior $q(\mu_p) = \delta(\mu_p)$ (MAP estimate) by optimizing the following ELBO lower bounding $\log p(\mathcal{D}, \mu_p)$:

$$\mathcal{L}(\mu_q, \sigma_q^2, \mu_p, \sigma_p^2) = \mathcal{L}_{data}(\mu_q, \sigma_q^2) - \frac{1}{2}\left[\frac{(\mu_q - \mu_p)^2 + \sigma_q^2}{\gamma} - 1 + \log\frac{\gamma}{\sigma_q^2}\right] - \frac{\mu_p^2}{2\alpha}. \tag{11}$$

we note that only the second and third term on RHS depends on $\mu_p$. Differentiating w.r.t. $\mu_p$ gives:

$$\frac{\partial}{\partial \mu_p}\mathcal{L}(\mu_q, \sigma_q^2, \mu_p, \sigma_p^2) = \frac{\mu_q - \mu_p}{\gamma} - \frac{\mu_p}{\alpha} = \frac{\alpha\mu_q - \alpha\mu_p - \gamma\mu_p}{\alpha\gamma}. \tag{12}$$

Setting the above equation to zero gives $\mu_p = \frac{\alpha}{\alpha+\gamma}\mu_q$. Next we calculate the coordinate ascend update:

$$\partial_{\mu_q}\mathcal{L}(\mu_q, \sigma_q^2, \mu_p, \sigma_p^2) = \partial_{\mu_q}\mathcal{L}_{data}(\mu_q, \sigma_q^2) - \frac{1}{\gamma}(\mu_q - \mu_p)|_{\mu_p=\frac{\alpha}{\alpha+\gamma}\mu_q} =$$

$$\mathcal{L}_{data}(\mu_q, \sigma_q^2) - \frac{\gamma}{\gamma(\alpha+\gamma)}\mu_q. \tag{13}$$

Hence coordinate ascend with MAP learning gives the same update as applying VI to learn $q(\mu_p)$. However, MAP learning neglects the constants allowing to learn $\alpha_{reg}$ by optimizing ELBO.

## B  Derivations

We first present detailed derivation for the bound in Eq. (9).

**Step I:** Recall that we assume fixed prior variances and the following priors $p(\boldsymbol{W}|\boldsymbol{\mu}_p) = \mathcal{N}(\boldsymbol{W}|\boldsymbol{\mu}_p, \gamma\mathbf{1})$, $p(\boldsymbol{\mu}_p) = \mathcal{N}(\boldsymbol{\mu}_p|\mathbf{0}, \alpha\mathbf{1})$, and posteriors $q(\boldsymbol{W}|\boldsymbol{\mu}_q, \boldsymbol{\sigma}_q^2) = \mathcal{N}(\boldsymbol{W}|\boldsymbol{\mu}_q, \boldsymbol{\sigma}_q^2)$, $q(\boldsymbol{\mu}_p) = \mathcal{N}(\boldsymbol{\mu}_p|\boldsymbol{m}, \boldsymbol{\beta})$. Since both prior and posterior follow mean-field assumption, we present derivations for an individual coordinate and add them up at the end.

**Step II:** To derive $\Phi$ in Eq. (6) we first calculate KL divergence between posterior and prior over $\mu_p$:

$$D_{KL}(q(\mu_p)||p(\mu_p)) = \frac{1}{2}\left[\frac{m^2 + \beta}{\alpha} - \log\beta + \log\alpha - 1\right]. \tag{14}$$

We have that $D_{KL}[q, p] = \mathcal{H}[q, p] - \mathcal{H}[q]$, So $\mathbb{E}_q \log p = -\mathcal{H}[q, p] = -D_{KL}[q, p] - \mathcal{H}[q]$. For $q(w|\mu_q, \sigma_q^2) = \mathcal{N}(w|\mu_q, \sigma_q^2)$, we have $\mathcal{H}[q] = \frac{1}{2}\log\sigma_q^2 + \log\sqrt{2\pi e}$, so first term in Eq. (6) is:

$$\mathbb{E}_{q(w|\mu_q, \sigma_q^2)}\log p(w|\mu_p, \sigma_p^2) =$$

$$-\frac{1}{2}\left[\frac{\sigma_q^2 + (\mu_q - \mu_p)^2}{\gamma} + \log\gamma - \log\sigma_q^2 - 1\right] - \frac{1}{2}\log\sigma_q^2 - \frac{1}{2}\log 2\pi e = \tag{15}$$

$$-\frac{1}{2}\left[\frac{\sigma_q^2 + (\mu_q - \mu_p)^2}{\gamma} + \log\gamma + \log 2\pi e - 1\right]. \tag{16}$$

Hence substituting into the definition of $\Phi(\mu_q, \sigma_q^2, q(\mu_p, \sigma_p^2))$:

$$\Phi(\mu_q, \sigma_q^2, q(\mu_p, \sigma_p^2)) = \mathbb{E}_{q(\mu_p, \sigma_p^2)}\left[\mathbb{E}_{q(w|\mu_q, \sigma_q^2)}\log p(w|\mu_p, \sigma_p^2)\right] - D_{KL}(q(\mu_p, \sigma_p^2)||p(\mu_p)p(\sigma_p^2)), \tag{17}$$

results in the expression:

$$\Phi = -\mathbb{E}_{q(\mu_p)}\frac{1}{2}\left[\frac{\sigma_q^2 + (\mu_q - \mu_p)^2}{\gamma} + \log\gamma + \log 2\pi e - 1\right] - \frac{1}{2}\left[\frac{m^2}{\alpha} + \frac{\beta}{\alpha} + \log\alpha - \log\beta - 1\right] =$$

$$-\frac{1}{2}\left[\mathbb{E}_{q(\mu_p)}\frac{\sigma_q^2 + (\mu_q - \mu_p)^2}{\gamma} + \log\gamma + \log 2\pi e + \frac{m^2}{\alpha} + \frac{\beta}{\alpha} + \log\alpha - \log\beta - 2\right]. \tag{18}$$

This expression has to be maximized w.r.t. $m$ and $\beta$ to derive $q^*$ which is used to form $\Phi^*$. However, using Eq. (7) gives the maximizer without needing to optimize for $m$:

$$\log q^*(\mu_p) \propto -\frac{(\mu_q - \mu_p)^2}{2\gamma} - \frac{\mu_p^2}{2\alpha} = -\frac{\alpha(\mu_q - \mu_p)^2}{2\alpha\gamma} - \frac{\gamma\mu_p^2}{2\alpha\gamma}$$

$$-\frac{\alpha\mu_p^2 + \alpha\mu_q^2 - 2\alpha\mu_p\mu_q + \gamma\mu_p^2}{2\alpha\gamma} = \frac{(\alpha+\gamma)(\mu_p - \frac{\alpha}{\alpha+\gamma}\mu_q)^2 + (\alpha - \frac{\alpha^2}{(\alpha+\gamma)^2})\mu_q^2}{2\alpha\gamma} = \tag{19}$$

$$\frac{(\mu_p - \frac{\alpha}{\alpha+\gamma}\mu_q)^2 + (\alpha - \frac{\alpha^2}{(\alpha+\gamma)^2})\mu_q^2}{2\frac{\alpha\gamma}{\alpha+\gamma}}. \tag{20}$$

This is Gaussian distribution, so $q^*(\mu_p) = \mathcal{N}(\mu_p | \frac{\alpha}{\alpha+\gamma}\mu_q, \frac{\alpha\gamma}{\alpha+\gamma})$.

**Step III:** We integrate $q^*(\mu_p)$ in the derived expression for $\Phi$:

$$\mathbb{E}_{\mathcal{N}(\mu_p | \frac{\alpha}{\alpha+\gamma}\mu_q, \frac{\alpha\gamma}{\alpha+\gamma})} \frac{(\mu_q - \mu_p)^2}{\gamma} = \tag{21}$$

$$\mathbb{E}_{\mathcal{N}(\mu_p | \frac{\alpha}{\alpha+\gamma}\mu_q, \frac{\alpha\gamma}{\alpha+\gamma})} \frac{\mu_q^2 + \mu_p^2 - 2\mu_p\mu_q}{\gamma} = \tag{22}$$

$$\frac{\mu_q^2 + (\frac{\alpha}{\alpha+\gamma}\mu_q)^2 + \frac{\alpha\gamma}{\alpha+\gamma} - 2\frac{\alpha}{\alpha+\gamma}\mu_q^2}{\gamma} = \frac{(\mu_q - \frac{\alpha}{\alpha+\gamma}\mu_q)^2 + \frac{\alpha\gamma}{\alpha+\gamma}}{\gamma} = \tag{23}$$

$$\frac{\frac{\gamma^2}{(\alpha+\gamma)^2}\mu_q^2}{\gamma} + \frac{\alpha}{\alpha+\gamma} = \frac{\gamma}{(\alpha+\gamma)^2}\mu_q^2 + \frac{\alpha}{\alpha+\gamma}. \tag{24}$$

Substituting $q^*(\mu_p)$ back into $\Phi$ and plugging in optimal $m$ and $\beta$ gives:

$$\Phi = -\frac{1}{2}\Big[\frac{\gamma}{(\alpha+\gamma)^2}\mu_q^2 + \frac{\alpha}{\alpha+\gamma} + \frac{\sigma_q^2}{\gamma} + \log\gamma + \log 2\pi e +$$

$$\frac{(\frac{\alpha}{\alpha+\gamma}\mu_q)^2}{\alpha} + \frac{\frac{\alpha\gamma}{\alpha+\gamma}}{\alpha} + \log\alpha - \log\frac{\alpha\gamma}{\alpha+\gamma} - 2\Big] = \tag{25}$$

$$-\frac{1}{2}\Big[\frac{\alpha+\gamma}{(\alpha+\gamma)^2}\mu_q^2 + \frac{\sigma_q^2}{\gamma} + \log\gamma + 1 - \log\frac{\gamma}{\alpha+\gamma} + \log 2\pi e - 2\Big] = \tag{26}$$

$$-\frac{1}{2}\Big[\frac{1}{\alpha+\gamma}\mu_q^2 + \frac{\sigma_q^2}{\gamma} + \log\gamma - \log\frac{\gamma}{\alpha+\gamma} + \log 2\pi e - 1\Big] = \tag{27}$$

$$-\frac{1}{2}\Big[\frac{\alpha_{reg}\mu_q^2 + \sigma_q^2}{\gamma} + \log\gamma - \log\alpha_{reg} + \log 2\pi e - 1\Big]. \tag{28}$$

**Step IV:** The full expression for the bound $\mathcal{L}_m^*$ becomes (we add $\mathcal{H}[q]$)

$$\mathcal{L}_{data}(\boldsymbol{\mu}_q, \boldsymbol{\sigma}_q^2) - \frac{1}{2}\sum_d \Big[\frac{\alpha_{reg}\mu_{q,d}^2 + \sigma_{q,d}^2}{\gamma} + \log\gamma - \log\alpha_{reg} - \log\sigma_{q,d}^2 - 1\Big]. \tag{29}$$

We now consider the derivation of the bound $\mathcal{L}_{mv}^*$ in Eq. (10). These calculations are similar in spirit to well-known derivations for conjugate priors in Bayesian hierarchical models.

**Step I:** We employ priors $p(w) = \mathcal{N}(w | \mu_p, \frac{1}{\tau_p})$, $p(\mu_p | \tau_p) = \mathcal{N}(\mu_p | 0, \frac{1}{t\tau_p})$ and $p(\tau_p) = \mathcal{G}(\tau_p | \alpha, \beta)$ and posteriors $q(w) = \mathcal{N}(w | \mu_q, \sigma_q^2)$, $q(\mu_p | \tau_p) = \mathcal{N}(\mu_p)$, $q(\tau_p) = \mathcal{G}(\tau_p)$.

**Step II:** We again use $\mathbb{E}_q \log p = -\mathcal{H}[q, p] = -D_{KL}[q, p] - \mathcal{H}[q]$ to derive:

$$\mathbb{E}_{q(w | \mu_q, \sigma_q^2)} \log p(w | \mu_p, \sigma_p^2) =$$

$$-\frac{1}{2}\Big[\frac{\sigma_q^2 + (\mu_p - \mu_q)^2)}{\sigma_p^2} + \log\sigma_p^2 + \log 2\pi e - 1\Big] = \tag{30}$$

$$-\frac{\tau_p}{2}(\sigma_q^2 + (\mu_p - \mu_q)^2) + \frac{1}{2}\log\tau_p - \log\sqrt{2\pi e} + \frac{1}{2}. \tag{31}$$

We first find the posterior for conditional distribution $\mu_p|\tau_p$. We have that $\log p(\mu_p|\tau_p) \propto -\frac{t\tau_p\mu_p^2}{2}$. We substitute into Eq. (7) to derive:

$$\log q^*(\mu_p|\tau_p) \propto -\frac{\tau_p}{2}(\sigma_q^2 + (\mu_p - \mu_q)^2) - \frac{1}{2}t\tau_p\mu_p^2 = \tag{32}$$

$$-\frac{\tau_p}{2}(\sigma_q^2 + \mu_p^2 + \mu_q^2 - 2\mu_q\mu_p + t\mu_p^2) \triangleq \tag{33}$$

$$-\frac{\tau_p}{2}\mu_p^2 + \tau_p\mu_q\mu_p - \frac{1}{2}t\tau_p\mu_p^2 = \tag{34}$$

$$-\frac{(1+t)\tau_p}{2}\mu_p^2 + \tau_p\mu_q\mu_p \triangleq \tag{35}$$

$$-\frac{(1+t)\tau_p}{2}(\mu_p^2 - \frac{2}{(1+t)}\mu_q\mu_p) \propto \log\mathcal{N}(\mu_p|\frac{1}{1+t}\mu_q, \frac{1}{(1+t)\tau_p}). \tag{36}$$

We now calculate the posterior for $\tau_p$. We derive $q^*(\tau_p)$ by marginalizing $\mu_p$ from the joint density $q^*(\tau_p, \mu_p)$. We have that $\log p(\tau_p) \propto (\alpha - 1)\log\tau_p - \beta\tau_p$ and $\log p(\mu_p) \propto -\frac{1}{2}t\tau_p\mu_p^2 + \frac{1}{2}\log\tau_p$. The density $q^*(\tau_p, \mu_p)$ satisfies (by again substituting into Eq. (7)):

$$\log q^*(\tau_p, \mu_p) \propto$$

$$-\frac{\tau_p}{2}(\sigma_q^2 + (\mu_p - \mu_q)^2) + \frac{1}{2}\log\tau_p - \frac{1}{2}t\tau_p\mu_p^2 + \frac{1}{2}\log\tau_p + (\alpha - 1)\log\tau_p - \beta\tau_p \triangleq \tag{37}$$

$$-\frac{\tau_p}{2}(\sigma_q^2 + 2\beta) + \alpha\log\tau_p - \frac{\tau_p}{2}(t\mu_p^2 + (\mu_p - \mu_q)^2). \tag{38}$$

To marginalize $\mu_p$ we need to integrate out $\exp[-\frac{\tau_p}{2}(t\mu_p^2 + (\mu_p - \mu_q)^2)]$:

$$\int \exp(-\frac{\tau_p}{2}(t\mu_p^2 + (\mu_p - \mu_q)^2))d\mu_p = \int \exp(-\frac{\tau_p}{2}(t\mu_p^2 + \mu_p^2 + \mu_q^2 - 2\mu_p\mu_q))d\mu_p = \tag{39}$$

$$\int \exp(-\frac{\tau_p}{2}((1+t)\mu_p^2 + \mu_q^2 - 2\frac{1+t}{1+t}\mu_p\mu_q))d\mu_p = \tag{40}$$

$$\int \exp(-\frac{\tau_p(1+t)}{2}(\mu_p^2 + \frac{1}{1+t}\mu_q^2 - 2\frac{1}{1+t}\mu_p\mu_q))d\mu_p = \tag{41}$$

$$\int \exp(-\frac{\tau_p(1+t)}{2}(\mu_p^2 + \frac{1}{(1+t)^2}\mu_q^2 - 2\frac{1}{1+t}\mu_p\mu_q - \frac{1}{(1+t)^2}\mu_q^2 + \frac{1}{1+t}\mu_q^2)d\mu_p = \tag{42}$$

$$\int \exp(-\frac{\tau_p(1+t)}{2}((\mu_p - \frac{1}{1+t}\mu_q)^2 - \frac{1}{(1+t)^2}\mu_q^2 + \frac{1}{(1+t)}\mu_q^2)d\mu_p = \tag{43}$$

$$\int \exp(-\frac{\tau_p(1+t)}{2}(\mu_p - \frac{1}{1+t}\mu_q)^2 + \frac{\tau_p}{2}\frac{1}{(1+t)}\mu_q^2 - \frac{\tau_p}{2}\mu_q^2)d\mu_p = \tag{44}$$

$$\int \exp(-\frac{\tau_p(1+t)}{2}(\mu_p - \frac{1}{1+t}\mu_q)^2 - \frac{\tau_p}{2}\frac{t}{(1+t)}\mu_q^2)d\mu_p = \tag{45}$$

$$\exp(-\frac{t}{2(1+t)}\mu_q^2\tau_p)\frac{\sqrt{2\pi}}{\sqrt{\tau_p(1+t)}}. \tag{46}$$

So after integrating out $\mu_p$ we derive (by substituting $-\frac{t}{2(1+t)}\mu_q^2\tau_p + \log\frac{\sqrt{2\pi}}{\sqrt{\tau_p(1+t)}}$ into the joint density we obtain:

$$\log q^*(\tau_p) \propto -\frac{\tau_p}{2}(\sigma_q^2 + 2\beta) + \alpha\log\tau_p - \frac{t}{2(1+t)}\mu_q^2\tau_p - \frac{1}{2}\log\tau_p = \tag{47}$$

$$\log q^*(\tau_p) \propto -\frac{\tau_p}{2}(\sigma_q^2 + 2\beta) + (\alpha - \frac{1}{2})\log\tau_p - \frac{t}{2(1+t)}\mu_q^2\tau_p \tag{48}$$

$$\log q^*(\tau_p) \propto -\frac{\tau_p}{2}(\sigma_q^2 + 2\beta + \frac{t}{1+t}\mu_q^2) + (\alpha - \frac{1}{2})\log\tau_p \tag{49}$$

$$\propto \log\mathcal{G}(\alpha + \frac{1}{2}, \beta + \frac{1}{2}(\sigma_q^2 + \frac{t}{(1+t)}\mu_q^2)). \tag{50}$$

Hence the $q^*$ are $q^*(\mu_p|\tau_p) = \mathcal{N}(\frac{1}{1+t}\mu_q, \frac{1}{(1+t)\tau_p})$ and $q^*(\tau_p) = \mathcal{G}(\alpha + \frac{1}{2}, \beta + \frac{1}{2}(\sigma_q^2 + \frac{t}{(1+t)}\mu_q^2))$.

**Step III:** We follow by forming the expression for $\Phi^*$ and marginalizing out prior parameters. First, to form $\Phi^*$, we need to calculate KL divergence between derived $q^*$ and corresponding priors. We now calculate KL divergence $D_{KL}(q^*(\mu_p|\tau_p)||\mathcal{N}(\mu_p|0, \frac{1}{t\tau_p}))$:

$$D_{KL}(\mathcal{N}(\mu_p|\frac{1}{1+t}\mu_q, \frac{1}{(1+t)\tau_p})||\mathcal{N}(\mu_p|0, \frac{1}{t\tau_p})) =$$

$$\frac{1}{2}\left[t\tau_p\left(\frac{1}{(1+t)^2}\mu_q^2 + \frac{1}{(1+t)\tau_p}\right) - \log\frac{t\tau_p}{(1+t)\tau_p} - 1\right] = \tag{51}$$

$$\frac{1}{2}\left[t\tau_p\frac{1}{(1+t)^2}\mu_q^2 + \frac{t}{(1+t)} - \log\frac{t}{1+t} - 1\right] = \tag{52}$$

$$\frac{1}{2}\left[\frac{t}{(1+t)^2}\tau_p\mu_q^2 + \frac{t}{(1+t)} - \log\frac{t}{1+t} - 1\right], \tag{53}$$

and recall the formula KL between Gamma distributions

$$D_{KL}(\mathcal{G}(x|\alpha_p, \beta_p)||\mathcal{G}(x|\alpha_q, \beta_q)) = (\alpha_p - \alpha_q)\psi(\alpha_p) - \log\Gamma(\alpha_p) +$$

$$+ \log\Gamma(\alpha_q) + \alpha_q(\log\beta_p - \log\beta_q) + \alpha_p\frac{\beta_q - \beta_p}{\beta_p}, \tag{54}$$

where $\psi$ denotes digamma function and $\Gamma$ denotes gamma function. Substituting posterior $\mathcal{G}(\sigma_p^2|\alpha + \frac{1}{2}, \beta + \frac{t}{2(1+t)}\mu_q^2 + \frac{1}{2}\sigma_q^2)$ and prior $\mathcal{G}(\sigma_p^2|\alpha, \beta)$ into the above equation gives:

$$D_{KL}(\mathcal{G}(\sigma_p^2|\alpha + \frac{1}{2}, \beta + \frac{t}{2(1+t)}\mu_q^2 + \frac{1}{2}\sigma_q^2)||\mathcal{G}(\sigma_p^2|\alpha, \beta)) =$$

$$= \frac{1}{2}\psi(\alpha + \frac{1}{2}) - \log\Gamma(\alpha + \frac{1}{2}) + \log\Gamma(\alpha) + \alpha\log[\beta + \frac{t}{2(1+t)}\mu_q^2 + \frac{1}{2}\sigma_q^2]$$

$$- \alpha\log\beta + \frac{(\alpha + \frac{1}{2})\beta}{\beta + \frac{t}{2(1+t)}\mu_q^2 + \frac{1}{2}\sigma_q^2} - \alpha. \tag{55}$$

Hence, to simplify $\Phi^*$ we need to integrate the expectations in the expression:

$$\Phi^* = \mathbb{E}_{q^*(\tau_p)}\mathbb{E}_{q^*(\mu_p)}\left[-\frac{1}{2}\left(\tau_p(\sigma_q^2 + (\mu_p - \mu_q)^2) - \log\tau_p + \log 2\pi e - 1\right)\right.$$

$$-D_{KL}(\mathcal{N}(\mu_p|\frac{1}{1+t}\mu_q, \frac{1}{(1+t)\tau_p})||\mathcal{N}(\mu_p|0, \frac{1}{t\tau_p}))\Big]$$

$$-D_{KL}(\mathcal{G}(\tau_p|\alpha + \frac{1}{2}, \beta + \frac{t}{2(1+t)}\mu_q^2 + \frac{1}{2}\sigma_q^2)||\mathcal{G}(\tau_p|\alpha, \beta)). \tag{56}$$

We divide the above expression and first integrate out $q^*(\mu_p)$ denoting it as $\star$.

$$\star = \mathbb{E}_{q^*(\mu_p)}\left[-\frac{1}{2}\left(\tau_p(\sigma_q^2 + (\mu_p - \mu_q)^2) - \log\tau_p + \log 2\pi e - 1\right)\right.$$

$$-D_{KL}(\mathcal{N}(\mu_p|\frac{1}{1+t}\mu_q, \frac{1}{(1+t)\tau_p})||\mathcal{N}(\mu_p|0, \frac{1}{t\tau_p}))\Big] = \tag{57}$$

$$\mathbb{E}_{q^*(\mu_p)} - \frac{1}{2}\left(\tau_p(\sigma_q^2 + (\mu_p - \mu_q)^2) - \log\tau_p + \log 2\pi e - 1\right)$$

$$- \frac{1}{2}\left[\frac{t}{(1+t)^2}\tau_p\mu_q^2 + \frac{t}{(1+t)} - \log\frac{t}{1+t} - 1\right]. \tag{58}$$

We have that

$$\mathbb{E}_{q^*(\mu_p)}(\mu_p - \mu_q)^2 = \mathbb{E}_{q^*(\mu_p)}\mu_p^2 + \mu_q^2 - 2\mu_q\mu_p = \tag{59}$$

$$\frac{1}{(1+t)^2}\mu_q^2 + \frac{1}{(1+t)\tau_p} + \mu_q^2 - \frac{2}{1+t}\mu_q^2 = \tag{60}$$

$$\frac{(1+t)^2 + 1 - 2(1+t)}{(1+t)^2}\mu_q^2 + \frac{1}{(1+t)\tau_p} = \tag{61}$$

$$\frac{t^2}{(1+t)^2}\mu_q^2 + \frac{1}{(1+t)\tau_p}. \tag{62}$$

So by substituting into the definition of $\star$ we have:

$$\star = -\frac{1}{2}\Big(\tau_p(\sigma_q^2 + \frac{t^2}{(1+t)^2}\mu_q^2 + \frac{1}{(1+t)\tau_p}) - \log \tau_p + \log 2\pi e - 1\Big) - \tag{63}$$

$$\frac{1}{2}\Big[\frac{t}{(1+t)^2}\tau_p\mu_q^2 + \frac{t}{(1+t)} - \log\frac{t}{1+t} - 1\Big] =$$

$$-\frac{\tau_p}{2}(\sigma_q^2 + \frac{t^2}{(1+t)^2}\mu_q^2) - \frac{1}{2(1+t)} + \frac{1}{2}\log \tau_p + \frac{1}{2} - \log\sqrt{2\pi e} -$$

$$\frac{1}{2}\frac{t}{(1+t)^2}\tau_p\mu_q^2 - \frac{1}{2}\frac{t}{(1+t)} + \frac{1}{2}\log\frac{t}{1+t} + \frac{1}{2} \tag{64}$$

$$= -\frac{\tau_p}{2}(\sigma_q^2 + \frac{t}{1+t}\mu_q^2) - \log\sqrt{2\pi e} + \frac{1}{2}\log \tau_p + \frac{1}{2}\log\frac{t}{1+t} + \frac{1}{2}. \tag{65}$$

We next need to integrate $\star$ w.r.t. $q^*(\tau_p)$ to derive $\Phi^*$. We use the fact that for Gamma distribution $\mathcal{G}(x|\alpha,\beta)$ we have that $\mathbb{E}\log X = \psi(\alpha) - \log \beta$, where $\psi$ denotes digamma function and $\mathbb{E}X = \frac{\alpha}{\beta}$. So we have that:

$$\mathbb{E}_{q^*(\tau_p)}\tau_p = \frac{\alpha + \frac{1}{2}}{\beta + \frac{1}{2}(\sigma_q^2 + \frac{t}{1+t}\mu_q^2)}, \tag{66}$$

$$\mathbb{E}_{q^*(\tau_p)}\log \tau_p = \psi(\alpha + \frac{1}{2}) - \log\Big[\beta + \frac{1}{2}(\sigma_q^2 + \frac{t}{1+t}\mu_q^2)\Big]. \tag{67}$$

We now integrate out approximate posterior $q^*(\tau_p)$:

$$\mathbb{E}_{q^*(\tau_p)}\star = \mathbb{E}_{q^*(\tau_p)}\Big[-\frac{\tau_p}{2}(\sigma_q^2 + \frac{t}{1+t}\mu_q^2) - \log\sqrt{2\pi e} + \frac{1}{2}\log \tau_p + \frac{1}{2}\log\frac{t}{1+t} + \frac{1}{2}\Big] = \tag{68}$$

$$-\frac{1}{2}\Big(\frac{\alpha + \frac{1}{2}}{\beta + \frac{1}{2}(\sigma_q^2 + \frac{t}{1+t}\mu_1^2))}\Big)(\sigma_q^2 + \frac{t}{1+t}\mu_q^2) - \log\sqrt{2\pi e} +$$

$$\frac{1}{2}\psi(\alpha + \frac{1}{2}) - \frac{1}{2}\log\Big[\beta + \frac{1}{2}(\sigma_q^2 + \frac{t}{1+t}\mu_1^2)\Big] + \frac{1}{2}\log\frac{t}{1+t} + \frac{1}{2}. \tag{69}$$

We next form $\Phi^*$:

$$\Phi^* = \mathbb{E}_{q^*(\tau_p)}\star - D_{KL}(\mathcal{G}(\tau_p|\alpha + \frac{1}{2}, \beta + \frac{t}{2(1+t)}\mu_q^2 + \frac{1}{2}\sigma_q^2)||\mathcal{G}(\tau_p|\alpha,\beta)) = \tag{70}$$

$$-\frac{\alpha + \frac{1}{2}}{2(\beta + \frac{t}{2(1+t)}\mu_q^2 + \frac{1}{2}\sigma_q^2)}(\sigma_q^2 + \frac{t}{(1+t)}\mu_q^2) - \frac{1}{2}\log[\beta + \frac{t}{2(1+t)}\mu_q^2 + \frac{1}{2}\sigma_q^2] - \log\sqrt{2\pi e}$$

$$+\frac{1}{2}\log\frac{t}{1+t} + \frac{1}{2} + \frac{1}{2}\psi(\alpha + \frac{1}{2}) - \Big[\frac{1}{2}\psi(\alpha + \frac{1}{2}) - \log\Gamma(\alpha + \frac{1}{2}) + \log\Gamma(\alpha)$$

$$+\alpha\log[\beta + \frac{t}{2(1+t)}\mu_q^2 + \frac{1}{2}\sigma_q^2] - \alpha\log\beta + \frac{(\alpha + \frac{1}{2})\beta}{\beta + \frac{t}{2(1+t)}\mu_q^2 + \frac{1}{2}\sigma_q^2} - \alpha\Big] = \tag{71}$$

$$= -\frac{\alpha + \frac{1}{2}}{2(\beta + \frac{t}{2(1+t)}\mu_q^2 + \frac{1}{2}\sigma_q^2)}(\sigma_q^2 + \frac{t}{(1+t)}\mu_q^2 + 2\beta) - (\alpha + \frac{1}{2})\log[\beta + \frac{t}{2(1+t)}\mu_q^2 + \frac{1}{2}\sigma_q^2] +$$

$$-\Big[-\log\Gamma(\alpha + \frac{1}{2}) + \log\Gamma(\alpha)\Big] - \alpha\log\beta - \alpha - \log\sqrt{2\pi e} + \log\frac{t}{1+t} + \frac{1}{2} = \tag{72}$$

$$-(\alpha + \frac{1}{2}) - (\alpha + \frac{1}{2})\log[\beta + \frac{t}{2(1+t)}\mu_q^2 + \frac{1}{2}\sigma_q^2] + \frac{\Gamma(\alpha + \frac{1}{2})}{\Gamma(\alpha)} + \alpha\log\beta + \alpha$$

$$-\log\sqrt{2\pi e} + \log\frac{t}{1+t} + \frac{1}{2}, \tag{73}$$

where in the main text we use $\delta = \frac{t}{1+t}$.

**Step IV:** We form the penalty in bound $\mathcal{L}^*_{mv}(\mu_q, \sigma_q^2)$ as :

$$\sum_d \left[ -(\alpha + \frac{1}{2}) \log[\beta + \frac{\delta\mu^2_{q,d}}{2} + \frac{1}{2}\sigma^2_{q,d}] + \frac{1}{2}\log\sigma^2_{q,d} + \log\frac{\Gamma(\alpha + \frac{1}{2})}{\Gamma(\alpha)} + \alpha\log\beta + \log\delta \right]. \quad (74)$$

## C  Experimental setup

Here we list all hyperparameters used in Section 4. We begin by vectorized MNIST experiment. For

| experiment | batch size | lr | epochs | init logvar | $\alpha_{reg}$ | $\alpha$ | $\beta$ | $\delta$ | $\gamma$ | test samples |
|---|---|---|---|---|---|---|---|---|---|---|
| MLP MNIST | 256 | 0.001 | 1000 | -7 | 0.05 | 0.5 | 0.01 | 0.1 | $\frac{20}{N_{in}}$ | 2000 |
| LeNet | 512 | 0.001 | 800 | -7 | 0.025 | 0.5 | 0.01 | 0.1 | $\frac{1}{N_{in}}$ | 500 |
| LeNet (CIFAR10/100, SVHN) | 512 | 0.001 | 800 | -7 | 0.2 | 0.5 | 0.01 | 0.5 | $\frac{1}{N_{in}}$ | 500 |
| CNN large | 512 | 0.001 | 800 | -7 | 0.025 | 0.5 | 0.01 | 0.5 | $\frac{1}{N_{in}}$ | 500 |
| CNN large (STL10) | 512 | 0.001 | 800 | -7 | 0.2 | 0.5 | 0.01 | 0.3 | $\frac{1}{N_{in}}$ | 500 |

Table 4: Hyperparameters used in our experiments.

LeNet and MNIST experiments, we do not use data augmentation. For larger CNN experiments, we use standard simple data augmenntation: padding in reflect mode with 4 (STL10 12) pixels and random cropping to initial size and random horizontal flip (exlcluding SVHN) with probability 0.5. For MNIST 1 epoch takes 1.2 sec, for LeNet 2.5 sec and for ResNet (CIFAR10) 1 epoch takes 12 sec.

## D  Additional experimental results

First we report results on UCI data sets as described in the main text but for 1 hidden layer BNN.

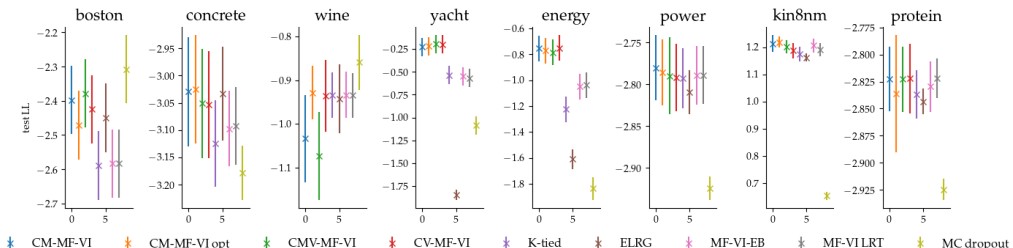

Figure 7: Comparison of collapsed bounds with other algorithms on UCI data sets (top) for 1 hidden layer BNN.

The collapsed bounds perform overall very well. They are outperformed by MC dropout on *boston* and *wine*, but MC dropout fails terribly on other data sets (we do not tune $p$ per data set, but use one setting we of $p$ working well for all data sets). We next report additional statistics for CNN experiments: test ECE score and test NLL on missclassified examples in the table below. We find that the collapsed bounds als perform well in these metrics, especially in ECE score.

We report the same statistics for large CNN experiment and additionally include final data terms from ELBO as they allow us to see if an algorithm is under-fitting/over-fitting. We again observe the collapsed bounds are performing well in terms of ECE score and, they have data terms which are relatively close to test NLLs reported in the main text (the gap between train NLL and test NLL is overall small and visibly smaller than for other algorithms).

| test NLL miss ↓ / ECE ↓ | MNIST | K-MNIST | F-MNIST | SVHN | CIFAR10 |
|---|---|---|---|---|---|
| CM-MF-VI | $1.63 \pm 0.09$ | $2.68 \pm 0.09$ | $2.05 \pm 0.10$ | $2.91 \pm 0.06$ | $2.27 \pm 0.01$ |
| CMV-MF-VI | $1.86 \pm 0.09$ | $2.85 \pm 0.13$ | $2.09 \pm 0.03$ | $2.77 \pm 0.02$ | $2.24 \pm 0.05$ |
| CV-MF-VI | $1.82 \pm 0.07$ | $2.75 \pm 0.09$ | $1.92 \pm 0.02$ | $2.66 \pm 0.02$ | $2.17 \pm 0.02$ |
| MF-VI | $1.69 \pm 0.01$ | $2.19 \pm 0.01$ | $1.84 \pm 0.05$ | $2.54 \pm 0.03$ | $2.09 \pm 0.02$ |
| MC dropout | $1.74 \pm 0.05$ | $2.49 \pm 0.07$ | $1.71 \pm 0.02$ | $2.14 \pm 0.00$ | $1.91 \pm 0.01$ |
| MAP | $2.37 \pm 0.10$ | $3.16 \pm 0.10$ | $2.04 \pm 0.03$ | $5.63 \pm 0.33$ | $2.90 \pm 0.21$ |
| CM-MF-VI | $0.00 \pm 0.00$ | $0.01 \pm 0.00$ | $0.01 \pm 0.00$ | $0.02 \pm 0.00$ | $0.02 \pm 0.00$ |
| CMV-MF-VI | $0.00 \pm 0.00$ | $0.01 \pm 0.00$ | $0.01 \pm 0.00$ | $0.01 \pm 0.00$ | $0.01 \pm 0.00$ |
| CV-MF-VI | $0.01 \pm 0.00$ | $0.01 \pm 0.00$ | $0.01 \pm 0.00$ | $0.02 \pm 0.00$ | $0.01 \pm 0.00$ |
| MF-VI | $0.02 \pm 0.00$ | $0.04 \pm 0.00$ | $0.03 \pm 0.00$ | $0.03 \pm 0.00$ | $0.04 \pm 0.01$ |
| MC dropout | $0.01 \pm 0.00$ | $0.02 \pm 0.00$ | $0.03 \pm 0.00$ | $0.10 \pm 0.00$ | $0.12 \pm 0.01$ |
| MAP | $0.00 \pm 0.00$ | $0.03 \pm 0.00$ | $0.01 \pm 0.00$ | $0.07 \pm 0.00$ | $0.12 \pm 0.02$ |

Table 5: Image classification with LeNet CNN comparison of test ECE and test NLL on missclassified examples.

| model | data set | CMV-MF-VI | CM-MF-VI | MF-VI | MC dropout | MAP |
|---|---|---|---|---|---|---|
| RESNET18 | STL10 | $1.91 \pm 0.02$ | $1.91 \pm 0.01$ | $1.85 \pm 0.01$ | $3.61 \pm 0.04$ | $5.47 \pm 0.12$ |
| | SVHN | $2.44 \pm 0.03$ | $2.55 \pm 0.01$ | $2.29 \pm 0.02$ | $2.34 \pm 0.03$ | $7.02 \pm 0.19$ |
| | CIFAR100 | $2.86 \pm 0.01$ | $3.25 \pm 0.03$ | $3.11 \pm 0.01$ | $3.19 \pm 0.01$ | $8.78 \pm 0.07$ |
| | CIFAR10 | $1.94 \pm 0.02$ | $2.09 \pm 0.01$ | $1.95 \pm 0.02$ | $1.97 \pm 0.01$ | $5.92 \pm 0.13$ |
| SHUFFLENET | STL10 | $1.89 \pm 0.02$ | $1.87 \pm 0.01$ | $1.81 \pm 0.01$ | $1.90 \pm 0.00$ | $4.10 \pm 0.11$ |
| | SVHN | $2.15 \pm 0.02$ | $2.21 \pm 0.01$ | $2.22 \pm 0.01$ | $1.99 \pm 0.00$ | $3.39 \pm 0.05$ |
| | CIFAR100 | $3.14 \pm 0.02$ | $3.23 \pm 0.03$ | $3.19 \pm 0.02$ | $3.56 \pm 0.01$ | $6.95 \pm 0.05$ |
| | CIFAR10 | $1.97 \pm 0.02$ | $1.97 \pm 0.02$ | $1.98 \pm 0.02$ | $1.90 \pm 0.01$ | $3.64 \pm 0.05$ |
| ALEXNET | STL10 | $1.84 \pm 0.00$ | $1.86 \pm 0.01$ | $1.92 \pm 0.03$ | $2.30 \pm 0.18$ | $4.85 \pm 1.00$ |
| | SVHN | $2.19 \pm 0.01$ | $2.27 \pm 0.03$ | $2.10 \pm 0.01$ | $2.02 \pm 0.02$ | $6.83 \pm 0.38$ |
| | CIFAR100 | $3.20 \pm 0.01$ | $3.28 \pm 0.03$ | $3.52 \pm 0.02$ | $3.26 \pm 0.02$ | $11.64 \pm 0.21$ |
| | CIFAR10 | $1.91 \pm 0.01$ | $1.97 \pm 0.01$ | $1.96 \pm 0.01$ | $1.90 \pm 0.01$ | $6.74 \pm 0.06$ |
| RESNET18 | STL10 | $1.07 \pm 0.01$ | $1.12 \pm 0.02$ | $1.66 \pm 0.01$ | $0.07 \pm 0.00$ | $0.03 \pm 0.00$ |
| | SVHN | $0.19 \pm 0.00$ | $0.13 \pm 0.00$ | $0.34 \pm 0.00$ | $0.20 \pm 0.00$ | $0.01 \pm 0.00$ |
| | CIFAR100 | $1.41 \pm 0.01$ | $0.72 \pm 0.01$ | $2.44 \pm 0.02$ | $0.99 \pm 0.00$ | $0.04 \pm 0.00$ |
| | CIFAR10 | $0.43 \pm 0.00$ | $0.28 \pm 0.00$ | $0.81 \pm 0.01$ | $0.39 \pm 0.00$ | $0.03 \pm 0.00$ |
| SHUFFLENET | STL10 | $1.02 \pm 0.02$ | $1.06 \pm 0.02$ | $1.75 \pm 0.07$ | $1.34 \pm 0.01$ | $0.24 \pm 0.00$ |
| | SVHN | $0.37 \pm 0.00$ | $0.36 \pm 0.00$ | $0.42 \pm 0.01$ | $1.47 \pm 0.01$ | $0.16 \pm 0.00$ |
| | CIFAR100 | $2.03 \pm 0.01$ | $1.66 \pm 0.04$ | $2.40 \pm 0.03$ | $3.30 \pm 0.01$ | $0.33 \pm 0.01$ |
| | CIFAR10 | $0.72 \pm 0.00$ | $0.71 \pm 0.01$ | $0.81 \pm 0.01$ | $1.41 \pm 0.00$ | $0.25 \pm 0.01$ |
| ALEXNET | STL10 | $1.65 \pm 0.05$ | $1.60 \pm 0.03$ | $1.98 \pm 0.02$ | $0.59 \pm 0.16$ | $0.19 \pm 0.11$ |
| | SVHN | $0.42 \pm 0.00$ | $0.34 \pm 0.01$ | $0.70 \pm 0.00$ | $0.53 \pm 0.01$ | $0.06 \pm 0.01$ |
| | CIFAR100 | $2.42 \pm 0.03$ | $2.19 \pm 0.10$ | $3.16 \pm 0.04$ | $1.98 \pm 0.03$ | $0.20 \pm 0.01$ |
| | CIFAR10 | $0.82 \pm 0.01$ | $0.72 \pm 0.03$ | $1.35 \pm 0.00$ | $0.74 \pm 0.03$ | $0.06 \pm 0.00$ |
| RESNET18 | STL10 | $0.07 \pm 0.01$ | $0.07 \pm 0.00$ | $0.04 \pm 0.00$ | $0.15 \pm 0.01$ | $0.22 \pm 0.00$ |
| | SVHN | $0.03 \pm 0.00$ | $0.02 \pm 0.00$ | $0.04 \pm 0.00$ | $0.04 \pm 0.00$ | $0.04 \pm 0.00$ |
| | CIFAR100 | $0.06 \pm 0.00$ | $0.02 \pm 0.00$ | $0.08 \pm 0.01$ | $0.04 \pm 0.00$ | $0.35 \pm 0.00$ |
| | CIFAR10 | $0.04 \pm 0.00$ | $0.02 \pm 0.00$ | $0.06 \pm 0.01$ | $0.04 \pm 0.00$ | $0.11 \pm 0.00$ |
| SHUFFLENET | STL10 | $0.07 \pm 0.01$ | $0.09 \pm 0.00$ | $0.04 \pm 0.01$ | $0.07 \pm 0.01$ | $0.25 \pm 0.01$ |
| | SVHN | $0.04 \pm 0.01$ | $0.04 \pm 0.00$ | $0.04 \pm 0.00$ | $0.31 \pm 0.01$ | $0.03 \pm 0.00$ |
| | CIFAR100 | $0.04 \pm 0.00$ | $0.02 \pm 0.00$ | $0.05 \pm 0.00$ | $0.11 \pm 0.00$ | $0.39 \pm 0.01$ |
| | CIFAR10 | $0.03 \pm 0.00$ | $0.03 \pm 0.00$ | $0.04 \pm 0.00$ | $0.13 \pm 0.00$ | $0.15 \pm 0.01$ |
| ALEXNET | STL10 | $0.11 \pm 0.02$ | $0.09 \pm 0.02$ | $0.03 \pm 0.01$ | $0.05 \pm 0.04$ | $0.22 \pm 0.03$ |
| | SVHN | $0.06 \pm 0.00$ | $0.05 \pm 0.01$ | $0.11 \pm 0.00$ | $0.11 \pm 0.01$ | $0.07 \pm 0.01$ |
| | CIFAR100 | $0.08 \pm 0.01$ | $0.08 \pm 0.00$ | $0.07 \pm 0.01$ | $0.09 \pm 0.00$ | $0.46 \pm 0.00$ |
| | CIFAR10 | $0.07 \pm 0.00$ | $0.06 \pm 0.01$ | $0.09 \pm 0.00$ | $0.09 \pm 0.00$ | $0.19 \pm 0.01$ |

Table 6: Test NLL on missclassified examples (top), data terms (middle) and test ECE score (bottom).

# E    Over-pruning

Here we provide more details on over-pruning in BNNs. First we consider a simple 1D regression task with 2 hidden layer BNN with a changing number of hidden units (size) with heteroscedastic Gaussian observation model. We plot contributions to predictions from 5 active units, remaining units (inactive) and predictions made by BNN (full pred). We observe that inactive units only contribute to homoscedastic predictive variance which grows with the size of the network. An increasing number of inactive weights introduce excessive noise to predictions, causing under-fitting when the size of the network grows further. These figures are in line with observations made in Figure 3.

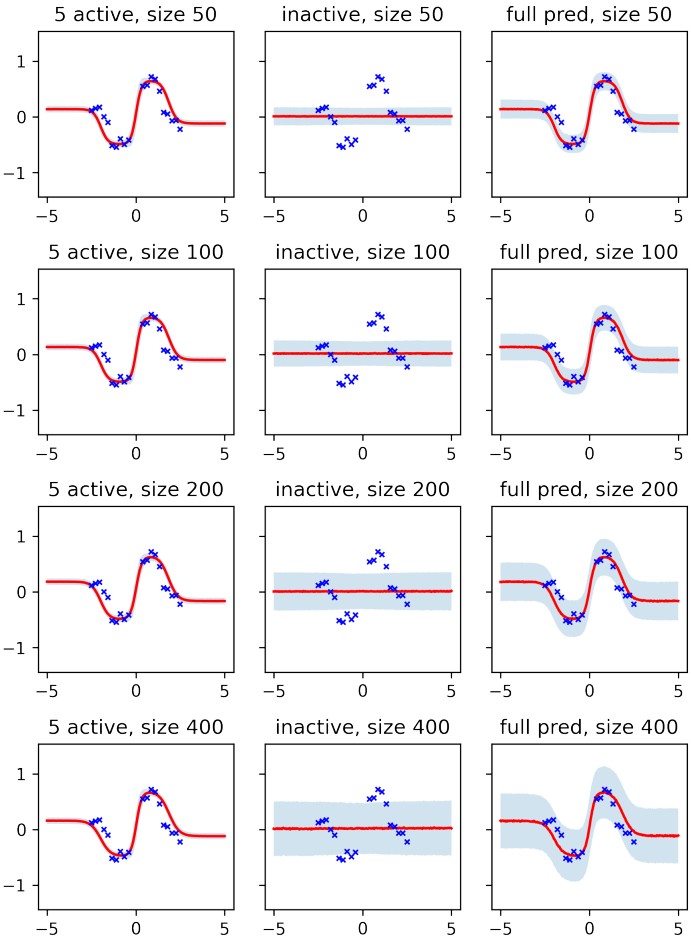

Figure 8: Over-pruning for 1D regression with 2 hidden layer BNN and *tanh* activations.

**Relation to heuristics.** To resolve under-fitting of mean-field BNNs, previous work has re-weighted terms arising from the KL divergence between the approximate posterior to improve the performance of variational BNNs. A generic expression to learn a variational posterior assuming prior $\mathcal{N}(\mathbf{0}, \sigma_p^2 \mathbf{1})$ can be expressed as:

$$\mathcal{L}_{heuristic}(\boldsymbol{\mu}_q, \boldsymbol{\sigma}_q^2) = \kappa_{data}\mathcal{L}_{data}(\boldsymbol{\mu}_q, \boldsymbol{\sigma}_q^2) - \mathbf{1}^T \left[ \frac{\kappa_1}{2\sigma_p^2}\boldsymbol{\sigma}_q^2 + \frac{\kappa_2}{2\sigma_p^2}\boldsymbol{\mu}_q^2 - \frac{\kappa_3}{2}\log\boldsymbol{\sigma}_q^2 + \frac{\kappa_4}{2}\log\boldsymbol{\sigma}_p^2 \right], \quad (75)$$

note that it is merely a heuristic and optimizing $\mathcal{L}_{heuristic}$ might not minimize a metric between approximate posterior $q(\boldsymbol{W}|\boldsymbol{\mu}_q, \boldsymbol{\sigma}_q^2)$ and true posterior $p(\boldsymbol{W}|\mathcal{D})$. Standard MF-VI corresponds to setting $\kappa_{data} = \kappa_1 = \kappa_2 = \kappa_3 = \kappa_4 = 1$. Posterior tempering [70] sets $\kappa_{data} = \kappa_1 = \kappa_2 = \kappa_3 = \frac{1}{T}$ and $\kappa_4 = 1$. KL down-weighting sets $\kappa_{data} = 1$, $\kappa_1 = \kappa_2 = \kappa_3 = \kappa_4 = \beta$. These approaches do not break the dependency $\kappa_2 = \kappa_1$ which we find to be inducing under-fitting. We have found one attempt to break the dependency $\kappa_2 = \kappa_1$ [45], but this was derived as a heuristic and resolving under-fitting has not been attributed to this change.