# OpenReview forum: "Collapsed Variational Bounds for Bayesian Neural Networks"
_NeurIPS.cc/2021/Conference — NeurIPS 2021 Poster_

### Official Review · Reviewer_KgBQ · 2021-07-08

**Rating:** 5
**Confidence:** 4

**Summary:**

This paper applies the idea of the collapsed variational bound studied by Teh et al., 2007 to BNNs. The bound is constructed by assuming that VB's Gaussian prior mean and variance are also part of the inference. (In contrast, one commonly sets them to some fixed values, e.g. $\mathcal{N}(0, 1)$.) The collapsed bound seems to have a similar computational cost but is tighter than the standard ELBO.

Extensive experiments show that the proposed objective works well, even on large networks, the regime which mean-field VB tends to underfit. It is also shown that the objective can mitigates the over-pruning effect that plagues VB.

**Limitations And Societal Impact:**

The authors discussed the limitations of their method in the paper.

**Main Review:**

I think the proposed method is interesting and is backed up by extensive experiments. This could be the answer to the difficulty of training VB due to hyperparameter selection. Nevertheless, I think the experiments can be expanded more, esp. regarding uncertainty quantification, OOD detection. Currently, the authors only show MNIST-FMNIST OOD detection with MLP. I think this is inadequate and the authors should use the standard OOD detection suites (MNIST, FMNIST, SVHN, CIFAR10, CIFAR100 in-distributions against 3+ OOD test sets) with a more commonly used network instead (ResNets, Wide-ResNets).

With that being said, the major issue with this paper is that it is *not* well-written. There is no paragraph break *at all* in this paper, making it a continuous blob of text. (Instead, the authors used \paragraph ``with its whitespace removed.) Maybe this due to some "LaTex hacks" that the authors used to make the text fit the page limit. If this is so, I suggest the authors not use them.
All in all, the presentation of this paper needs to be improved significantly and I urge the authors to do so meticulously.

**Questions:**
* Based on the discussion in L.80-84: How does the collapsed bound compare to EM?

**Suggestions:**
* Use paragraph break and use whitespace judiciously
* Use display math judiciously, instead of putting long formulas inline
* Put L.120-123 inside LaTex's algorithm env. Inline, it is very confusing to read
* Figures are very hard to read. I suggest the author to use tikz

**Minor suggestions:**
* L.46: "..., *the* ELBO lowers bound ...", also add a period before "Replacing ..."
* Fig. 3: Add y-label to the second row
* Tab. 1: What is "test ER"? (It's discussed in the next page, but it should be much closer to Tab. 1)
* Tab. 2, 3: Unclear which one is test-NLL, which one is test-ER.


---

**Post-Rebuttal**

While I like the idea, I still think that it would be beneficial for the paper to be given more careful treatment in its writing and presentation---this will ultimately change the paper substantially and thus, I believe, is more suitable for resubmission. My updated score (weak reject) reflects this ambivalence.

**Time Spent Reviewing:**

3

---

> ### Author Response · Authors · 2021-08-10
> **We thank the reviewer for the comments and time spent reviewing the paper.**
>
> We thank the reviewer for the comments and time spent reviewing the paper. We acknowledge that the writing can be improved, and we refer to the reviewer to our general comment for specific ways to improve and restructure the writing. This includes addressing the comments raised by the reviewer (using paragraph breaks, separating algorithms into a box and re-plotting the figures to make them larger/easier to read). As stated in the general comment we will also decompress the text to make it easier to read. Although our plan to improve the clarity of the writing is very specific, we welcome further comments that would help us to improve the clarity of presentation.
>
> At the same time, we believe the paper can still be understood to some extent in its current form, perhaps with more effort than it should be needed (reviewer A9Jx: “Because of its nature, the paper is quite dense from the theoretical point of view. In most parts derivations are well explained, but there are some places that require clarification”).
>
>
> Regarding how the collapsed bound compares to EM:
> We thank the reviewer for asking this and will include a mention in the Related Work section. We acknowledge there’s some similarity with variational version of EM [2]. In EM both updates are analytical but we learn variational parameters with gradient updates. Our work is more centered around employing specific hierarchical models, solving analytically for optimal distribution of hyperparameters and collapsing the bound. We end up with an optimization problem of only variational parameters (as opposed to two steps in EM). Our work is inspired more by making MacKay’s approach to learn hyperparameters in neural networks [3] practical by using collapsed bounds rather than EM algorithm itself.
>
> To clarify further, in variational EM the likelihood usually depends both on latent variables $z$ and model parameters $\theta$: $p(x,z| \theta)$, where in our case the likelihood does not depend on hyperparameters: $p(y|x, w)$ but only on weights $w$. We exploit this to derive analytical solutions for distributions over hyperparameters.  This allows us to collapse the bound (which cannot be done in general for EM) and reduce the problem to optimization of only variational parameters (we substitute optimal q’s for hyperparameters in ELBO). This is explained in more detail in section 3.1 of [1] (in the context of GPs).
>
> Regarding further OOD experiments: We can include additional OOD results in an appendix if the reviewer considers this crucial (ResNet + more data sets). The current focus of the experimental section is to demonstrate that our approach stops severe undefitting of MF-VI.
>
> [1] Miguel Lázaro-Gredilla and Michalis K. Titsias. 2011. Variational heteroscedastic Gaussian process regression. In Proceedings of the 28th International Conference on International Conference on Machine Learning (ICML'11). Omnipress, Madison, WI, USA, 841–848.
>
> [2] M. J. Beal and Z. Ghahramani, “The Variational Bayesian EM Algorithm for Incomplete Data: With Application to Scoring Graphical Model Structures”, Bayesian statistics, vol. 7
>
> [3] David J. C. MacKay. Comparison of approximate methods for handling hyperparameters. Neural Comput., 11(5):1035–1068, July 1999

---

> > ### Comment · Reviewer_KgBQ · 2021-08-25
> > **Thank you for your response**
> >
> > Thank you for your response. I applaud the authors for acknowledging the writing issue and provided a plan on how to fix this. While I like the idea, I still think that it would be beneficial for the paper to be given more careful treatment in its writing and presentation---this will ultimately change the paper substantially and thus, I believe, is more suitable for resubmission. My updated score (weak reject) reflects this ambivalence.
> >
> > A minor point about EM: EM doesn't have to be analytical. The main point is that at each iteration, it forms a variational approximation $q$ and uses the ELBO under $q$ to optimize hyperparameters---the way we do the approximation and the optimization are not important.

---

### Official Review · Reviewer_rzGC · 2021-07-13

**Rating:** 5
**Confidence:** 3

**Summary:**

Although marginal likelihood can be used to select hyper-parameters, the ELBO is generally regarded as too weak a bound to do this effectively.
The authors propose applying collapsed variational bounds to variational inference of BNNs.
They show why this maintains the tightness of the bound for parameter inference while also providing a tightish bound for hyper-parameter optimization by exploiting the independence of the two.
They empirically show that their method performs well in a range of settings, including in tests of ood uncertainty.

**Limitations And Societal Impact:**

These are adequately addressed.

**Main Review:**

# Overall review
I found this paper difficult to assign a score to, and I remain quite uncertain.
On the one hand, I believe that you have proposed a sensible approach which fixes a moderately important problem.
You have also provided extensive experimental support for your hypotheses.
On the other hand, I found the paper quite hard to read and somewhat poorly organized.
I feel that it would benefit a lot from being redrafted in a way that makes it clearer what its contributions are and how they are being examined.
For that reason I am currently recommending borderline rejection, but I could easily be persuaded upwards if other reviewers think particularly highly of the significance of the work.

# Major comments

There is a lot of redundancy.
The abstract could probably be pruned by a third.
Much of the introduction is relatively boiler-plate and does not focus on the specific contribution that *you* are making.
I did not quite understand what the role of the "Preliminaries" section was. Probably you could take a small part of this and place it at the start of what is now S3.
Similarly, I would reserve a discussion of over-pruning until the experiments section and make a note in the introduction that your method resolves an over-pruning issue in VI.
When I read it I had no idea why this was being mentioned and Figure 1 is somewhat a distraction at this stage.

I think some of the details of the derivations in S3 could be moved to an appendix, with more space devoted to an explanation of why you are doing things and what the intuitive purpose is and less on the details.

The captions on many figures seem incomplete, with only some subfigures discussed explicitly.
The figures are all far too small and difficult to read.

I would be inclined to pick your experiments more carefully for the main body so that you can devote more time/space to explaining and interpreting each thoroughly.
They feel rushed and crowded.

# Minor remarks

Regarding the introduction: I'm easily persuaded that a tighter bound to the marginal likelihood would be good, so a lot of this background isn't super necessary.
I would have liked more of a sketch of what your approach is going to be.

Please make your references hyperlinks.
Especially when you use a numerical citation style it is almost impossible to figure out what you are citing otherwise.

The spacing of the paper is very tight, perhaps some of the style-sheet was accidentally overridden?
This makes it much harder to read.

**Time Spent Reviewing:**

3

---

> ### Author Response · Authors · 2021-08-10
> **We thank the reviewer for the comments.**
>
> We thank the reviewer for the comments and apologize for the lack of clarity in writing. We much appreciate precise suggestions proposed by the reviewer. Using these suggestions as a backbone, we designed a specific plan to improve the clarity of writing which we included in a general comment. We hope this explicitly addresses many of your comments and suggestions.
>
> Regarding the Introduction: We tried to make the introduction/preliminary section more accessible to people less familiar with variational Bayes for BNNs (e.g. over-pruning or giving up on data), which might have resulted in a feeling some parts are redundant when read by an expert. We described the specific improvements based on the reviewer suggestion in the general comment. For example, we will merge the Preliminaries section with Section 3.
>
> Regarding the experiments: As suggested by the reviewer, we will explain the meaning and implications of our empirical findings in a better way instead of simply demonstrating the improvements in metrics. For instance, we will explain that using fixed prior implies over-pruning of weights, underfitting and poor train/test NLL. We will also explain that this leads to a situation where simpler variational posteriors give better predictive performance (as they cannot be fully pruned) [3]. We will also explain how our results showing improvements in predictive performance link to the relevant previous work discussing the performance of mean-field VI in BNNs e.g. [2]. If there’s not enough space, we’ll move some of the results to the Appendix. We note that we will be allowed another page for a camera-ready version of the paper.
>
> Regarding the abstract: We will prune and paraphrase the first two-three sentences of the abstract.
>
> We will also ensure the references appear as hyperlinks in the final version and spacings in the text are increased.
>
> [1] Coker, B., Pan, W., & Doshi-Velez, F. (2021). Wide Mean-Field Variational Bayesian Neural Networks Ignore the Data.
>
> [2] Sebastian Farquhar, Lewis Smith, Yarin Gal Liberty or Depth: Deep Bayesian Neural Nets Do Not Need Complex Weight Posterior Approximations
>
> [3] Brian L Trippe, Richard E. Turner, Overpruning in Variational Bayesian Neural Networks

---

> > ### Comment · Reviewer_rzGC · 2021-09-01
> > **Thanks for your response**
> >
> > I feel like you have a very good plan for how to improve the paper. Unfortunately, because you aren't able to upload a revised version at this stage, I don't feel I can increase my score because it would involve significant revisions.
> >
> > However, I also think that the paper is sufficiently interesting that it may well be worth accepting even with its current lack of readability. I'm genuinely unsure about whether this paper ought to be accepted and think that either outcome would be reasonable, and I've made this clear in discussion.

---

### Official Review · Reviewer_PHT4 · 2021-07-16

**Rating:** 6
**Confidence:** 2

**Summary:**

This work introduces an improvement over the typical ELBO by deriving a tighter bound to the log marginal likelihood. It further shows that the ELBO can be used to learn appropriate hyperparameters which can help in situations where MF-VI would have underfit. A few variants of this method are derived where some combination of prior means/variances are learned. These methods were shown to be competitive with and at times outperform existing methods in experiments on standard benchmarking datasets.

**Limitations And Societal Impact:**

The authors have not addressed this, but this decision is appropriate for the content.

**Main Review:**

Other work in this area has largely centered on improvements on gradients of the ELBO this work focuses on an improvement on the bound of the log marginal likelihood. These improved bounds are shown mathematically with sufficient detail. This work builds on existing ideas from Bayesian hierarchical models. The idea of learning hyperparameters seems to be related in spirit to empirical Bayes applied to neural networks. [1]

It is hard to read the algorithm as it is currently written in paragraph form; breaking this into its own section/some sort of indentation would help.. The provided example right after this was illustrative and helpful.

Just above equation (2) there is a missing period “.”: “0, ELBO lower bounds log marginal likelihood $\log p(D|µ_p,σ^2_p) ≥ L(µ_q,σ^2_q;µ_p,σ^2_p)$ Replacing” → “0, ELBO lower bounds log marginal likelihood $\log p(D|µp,σ2 p) ≥ L(µ_q,σ^2_q;µ_p,σ^2_p)$. Replacing”


[1] G. Casella, “An Introduction to Empirical Bayes Data Analysis,” The American Statistician, vol. 39, no. 2, pp. 83–87, 1985, doi: 10.2307/2682801.


----------------------
Thanks for the reply. After further discussion and thought I have lowered my rating to marginally above the acceptance threshold. The idea is good but the structure and writing is weak.



**Time Spent Reviewing:**

8

---

> ### Author Response · Authors · 2021-08-10
> **We thank the reviewer for their time spent reviewing the paper.**
>
> We thank the reviewer for their time spent reviewing the paper, and for detailed comments and appreciating the technical content of the paper.
>
> We will improve the clarity of writing as discussed in a general comment (this includes extracting the algorithm from text into an algorithm box). We’ll also correct pointed out typos.
>
> As the reviewer points out, our approach is connected to empirical Bayes (type II MLE) where we learn a distribution over hyperparameters as opposed to using point estimates.
> However, we use ELBO as an optimization target (this approach has been proposed by MacKay [1] and is a common practice for variational approximations to Gaussian Processes) as opposed to optimizing log marginal likelihood. We will clarify this in the Related Work section and include the mentioned citation.
>
> [1] David J. C. MacKay. Comparison of approximate methods for handling hyperparameters. Neural Comput., 11(5):1035–1068, July 1999.

---

### Official Review · Reviewer_A9Jx · 2021-07-16

**Rating:** 8
**Confidence:** 3

**Summary:**

This paper defines bayesian neural network models with hierarchical prior distributions. These models are trained efficiently using collapsed variational bounds, in which some of the variables of the hierarchical ELBO are integrated out. The authors show that these models outperform their hierarchical counterparts in a number of experiments and standard benchmarks.

**Limitations And Societal Impact:**

Yes

**Main Review:**

I found this paper to be a very interesting read, and a solid contribution to the fundamental research question on how to best model uncertainty in deep neural networks.

Using hyperpriors makes a lot of sense to me, since as it is also shown in the experiments, fixing the values of the hyperparameters as normally done is not a robust solution.
To the best of my knowledge, this is the first work that extends MF-VI for bayesian deep learning using ideas on collapsed bounds previously introduced for hierarchical bayesian models.
By using distribution from the exponential family the authors show a scalable and effective way to learn such models.

Because of its nature, the paper is quite dense from the theoretical point of view. In most parts derivations are well explained, but there are some places that require clarification:

1. line 87: "where for clarity we omit. the parameterization.." this actually caused me some confusion since I did not know why you would do this. You should say that you do it since it will be implicitly defined.

2. line 107: "straightforward to derive..." missing a reference for readers not familiar with MF-VI (Bishop's PRML book could be a good one for example)

3. line 146: why are there 2 different priors over W?

4. line 148: distribution G in p($\tau$) is not defined, i assume it is a Gamma distribution?

5. line 215: what is p? is it the dropout probability?

Overall, I believe this paper can have a big impact on the area of bayesian deep learning, so I vote for accepting it.


___________________________________________
Reply to author's rebuttal.
Thanks for your reply, I will leave the score unchanged and argue for acceptance.

**Time Spent Reviewing:**

4

---

> ### Author Response · Authors · 2021-08-10
> **We thank the reviewer for their time spent reviewing the paper.**
>
> We thank the reviewer for their time spent reviewing the paper, and for detailed comments and appreciating the technical content of the paper. We also believe that our method can have an important impact on the community: both in terms of warranting good predictive performance of variational Bayes methods applied to neural networks and understanding the reasons why having a fixed prior often fails in terms of predictive performance [1].
>
> We have also since found that our method not only gives better predictive performance, but also allows us to train BNNs using larger batches of data as our approach does not overfit/underfit. This drastically reduces training time. For instance, training on vectorized MNIST for 1000 epochs with batch size of 5000 requires 0.093s/epoch and results in test NLL 0.041 and test error rate 1.34% at convergence (better local optimum than using batch size 512 which results in test NLL 0.047 and the same test error rate but takes 0.83s per epoch). So in this case our algorithm provides better predictions and reduces the time required to train mean-field variational BNN by an order of magnitude.
>
>
> To clarify the specific points:
> 1. As suggested, we will say that the parametrization is implicit. We apologise for the confusion with the notation. The main reason for omission of variational parameters is that the equations are getting too long.
> 2. We will add the proposed citation, thank you for the suggestion.
> 3. This is a typo; there should be only one prior where both means and variances are inferred.
> 4. Yes, G denotes Gamma distribution. We will make this more clear.
> 5. Yes, $p$ is a dropout probability. We will mention this explicitly in the text.
>
> We also hope that our suggestions for re-structuring some of the paper content in the general response will ease the burden on the reader, making the paper a little less dense theoretically.
>
> [1] Coker, B., Pan, W., & Doshi-Velez, F. (2021). Wide Mean-Field Variational Bayesian Neural Networks Ignore the Data.

---

### Author Response · Authors · 2021-08-10
**We thank all the reviewers for their time and effort in reviewing the paper.**

We thank all the reviewers for their time and effort in reviewing their papers, and for their suggestions on how to improve it. We are glad that all reviewers agree that the theory in the paper is strong, novel and useful to the community. Reviewer A9Jx: “Overall, I believe this paper can have a big impact on the area of bayesian deep learning, so I vote for accepting it.”, Reviewer KgBQ “I think the proposed method is interesting and is backed up by extensive experiments.”, Reviewer rzGC “I believe that you have proposed a sensible approach which fixes a moderately important problem”.
We acknowledge the reviewers’ concerns regarding presentation and writing, and we aim to address this explicitly here. We also respond to each reviewer’s specific points in a separate response to each reviewer.

We promise to restructure the paper and propose a specific plan, based on the helpful guidance given by Reviewer rzGC. These changes focus purely on presentation and do not alter the theoretical contributions, empirical findings or core content of the paper. Unfortunately we cannot post a new version of the paper for the reviewers to see.

(1) We will separate out the algorithm for deriving tighter bounds (lines 119-123) into an algorithm box, as suggested by Reviewers rzGC and PHT4.

(2) Increase spacing between bold headers, and increase the size of figures and text in figures as suggested by Reviewer rzGC and KgBQ.

(3) Moving the paragraph and Figures on “Over-pruning in BNNs” (lines 59-73) to the Experiments section (after line 206), as suggested by Reviewer rzGC. We agree that this will free up space early on and streamline the paper.

(4) Rewrite/Merge the “Preliminaries” and “Collapsed Variational Bounds for Bayesian Neural Networks” sections into 3 subsections: (i) discussing the derivation of the method, (ii) algorithm, and (iii) examples and remove background information in lines 46-53 as suggested by Reviewer rzGC.

(5) When discussing the derivation of the method, we will move some equations into an appendix, and focus on the high-level picture and motivation. We will add a sentence explaining our approach in an intuitive way both to introduction (line 30) and when describing the algorithm again as suggested by Reviewer rzGC.

We also promise to open source the implementation of the algorithm to improve the clarity of the methods described in the paper.

---

### Decision · Program_Chairs · 2021-09-27

**Decision:**

Accept (Poster)

**Comment:**

This paper tightens the standard variational bound optimized in Bayesian Deep Learning by drawing on collapsed variational inference. The authors consider the prior parameters as latent variables and derive a hierarchical variational inference procedure in which the top-level latent variables are marginalized out. The authors showed strong empirical performance of their method compared to a variety of baselines.

The paper provides a non-trivial methodological contribution to the Bayesian deep learning community. Its mathematical derivations are not simple but generally well-explained (the authors sometimes don’t clearly distinguish between latent variables and variational parameters at times).  The main point of criticism was suboptimal structuring and writing. I strongly encourage the authors to make their paper more accessible by following the detailed advice that the reviewers provided. Overall, this is a very good paper.